# Heterogeneous nuclear ribonucleoprotein U (HNRNPU) safeguards the developing mouse cortex

Tamar Sapir[1], Aditya Kshirsagar [1], Anna Gorelik[1], Tsviya Olender[1], Ziv Porat [2], Ingrid E. Scheffer [3], David B. Goldstein[4], Orrin Devinsky [5] & Orly Reiner [1✉]

*HNRNPU* encodes the heterogeneous nuclear ribonucleoprotein U, which participates in RNA splicing and chromatin organization. Microdeletions in the 1q44 locus encompassing *HNRNPU* and other genes and point mutations in *HNRNPU* cause brain disorders, including early-onset seizures and severe intellectual disability. We aimed to understand HNRNPU's roles in the developing brain. Our work revealed that HNRNPU loss of function leads to rapid cell death of both postmitotic neurons and neural progenitors, with an apparent higher sensitivity of the latter. Further, expression and alternative splicing of multiple genes involved in cell survival, cell motility, and synapse formation are affected following *Hnrnpu's* conditional truncation. Finally, we identified pharmaceutical and genetic agents that can partially reverse the loss of cortical structures in *Hnrnpu* mutated embryonic brains, ameliorate radial neuronal migration defects and rescue cultured neural progenitors' cell death.

[1] Department of Molecular Genetics, Weizmann Institute of Science, Rehovot, Israel. [2] Flow Cytometry Unit, Life Sciences Core Facilities, Weizmann Institute of Science, Rehovot, Israel. [3] The University of Melbourne, Austin Health and Royal Children's Hospital, Florey and Murdoch Children's Research Institutes, Melbourne, VIC, Australia. [4] Institute for Genomic Medicine, Columbia University, New York, NY, USA. [5] NYU Langone Medical Center, NYU, New York, NY, USA. ✉email: orly.reiner@weizmann.ac.il

Elimination of cells is critical to maintaining tissue homeostasis and is vital for proper development. Cell death can result from various mechanisms of which TP53 is a crucial player via canonical and non-canonical pathways (reviewed by refs. [1,2]). Programmed cell death during mouse corticogenesis occurs during embryogenesis and postnatal development[3]. In the developing mouse, brain progenitors are mostly eliminated from the pallium. Mosaic analysis studies revealed asymmetric cell death after the first division that involved approximately 70% of the intermediate progenitors derived clones[4]. Increased cell death in the developing brain can result in pathologies, as have been documented in cases of mutations associated with autosomal recessive primary microcephaly[5]. These mutations delay mitotic progression and activate a TP53-dependent mitotic surveillance pathway[6]. The second wave of programmed cell death occurs during the first two weeks of life. It eliminates most of the transient populations, including Cajal-Retzius cells[7,8] and subplate neurons, and fine-tunes the density of other cortical populations[3]. The extent of cell death in the cortex is cell type and region-specific and negatively correlates with electrical activity[9]. It is essential for shaping cortical connectivity, maintaining a proper excitation/inhibition balance[10], and for functional myelination of interneurons[11].

The regulation of cell death-mediated pathways occurs at multiple stages, including transcriptional, post-transcriptional, and post-translational levels. RNA splicing is a post-transcriptional event that allows the inclusion or exclusion of sequences depending on specific cellular settings. The spliceosome mediates splicing, a complex containing RNA (e.g., small nuclear ribonucleoproteins and heterogeneous nuclear ribonucleoproteins (HNRNPs)) and other proteins. HNRNPU encodes for the heterogeneous nuclear ribonucleoprotein U[12], also known as scaffold attachment factor A, a nuclear chromatin organizer[13]. HNRNPU is critical in mammalian development, and knockout mice exhibit early lethality[14]. Microdeletions in the 1q44 locus encompassing HNRNPU and other genes, as well as point mutations in HNRNPU, result in brain disorders, including early-onset seizures, severe intellectual disability, and lower penetrance microcephaly, a thin corpus callosum, dysmorphic facial features, and hypotonia[15–21] (reviewed by ref. [22]). However, understanding of HNRNPU's functions in the developing brain is lacking. The HNRNPU protein binds to RNA, DNA, and other proteins, and these interactions facilitate its functions in organizing and stabilizing nuclear chromatin, regulating gene transcription[23–32], RNA splicing[33–35], and RNA stability[36–39]. HNRNPU displays a dynamic expression pattern in cultured cells and contributes to spindle assembly and stabilizing kinetochores-MT attachments[40,41].

Our work revealed that HNRNPU is required for the survival of cultured neural progenitors in vitro and of the entire mouse cortex in vivo. Removal of HNRNPU activates death programs that were under check. We identified genes whose expression levels vary and alternatively spliced genes when Hnrnpu is mutated. These genes participate in cell viability, cell motility, and other functions, causing the death of neural progenitors and abnormalities in neuronal migration. The p53, TGFB1, REST, and different critical pathways were affected. We suppressed cell death by pan-Caspase inhibitors, p53 inhibitors, and necroptosis inhibitors, demonstrating the involvement of p53-dependent canonical and noncanonical death mechanisms in neural progenitors. We showed that reducing the levels of the splicing factor SRSF3 opposes HNRNPU's loss of function effects and improves neural progenitors' viability and neuronal migration.

## Results

**HNRNPU is essential for cerebral cortex development.** The mouse developing brain expresses abundant levels of HNRNPU,

most prominent in the apical aspect of the ventricular zone (VZ) and the cortical plate (Fig. 1A-C'). Within the VZ, we detected a high HNRNPU signal in metaphase cells. In those dividing cells, HNRNPU occupies the space between the condensed chromosomes and the poles (Fig. 1A-A'). HNRNPU is also localized to centrosomes and cilia (colocalized with the cilia marker, acetylated tubulin (Fig. 1B-B'), and the centriole component CENP-B (Fig. 1C-C'). HNRNPU in dissociated neurospheres was analyzed using an ImageStream flow cytometer (Fig. 1D, E). The protein expression is dynamic, and while mitotic cells express high levels of HNRNPU, the protein is not attached to chromatin. Conversely, cells in G1/S display reduced HNRNPU signal intensity, which is better colocalized with chromatin.

To study HNRNPU's role in the developing brain, we used a conditional Hnrnpu allele[34] crossed the mice with Emx1Cre mice. The observed phenotype in the mutated brains (Emx1cre/+ Hnrnpu fl/fl) was dramatic. The brain regions in which Emx1Cre is expressed and Hnrnpu is mutated, were progressively omitted, whereas E18 cortices were smaller, and P8 brains lacked the telencephalon (Fig. 1F–I). At E18, the medial areas of the cortex are missing, but remanence of the cortex is still visible laterally (G, M). These areas express Tbr1, yet its typical localization at deeper cortical layers is lost (Fig. 1K, N). Astrocyte lineage can be detected in the mutant cortices; however, GFAP expression at the cortical plate is reduced by ~40% (Fig. 1L, O, P). At 21 days, the mutated animals survive, despite a smaller size and a substantial loss of all cortical structures. We could not anatomically annotate the "appendix" that is visible at this age (Fig. 1T, V); however, this region expresses an unorganized mixture of neuronal and glial markers (Fig. 1Q–V). Lineage tracing (E14-P21) of the progeny of electroporated neural progenitors with reduced HNRNPU levels revealed no bias toward reduction in the number of astrocytes at postnatal ages (P21, Supplementary Fig. 1).

We followed the survival of ex utero electroporated cultured cortical cells using time-lapse microscopy (Fig. 2A–M) to better understand this phenotype. The cultured cells were imaged for three hours (Fig. 2A–M) and fixed after two days in vitro (Fig. 2N–V). Hnrnpu fl/fl embryonic brains were electroporated with a dual-color Cre reporter, CAG::spotlight (CAG::SL), in combination with either no Cre, a constitutive Cre (CAG::NLS-Cre-GFP), intermediate progenitors Cre (Tbr2::CRE), or a postmitotic neuronal Cre (Ta1::Cre). The CAG spotlight constitutively expresses a green fluorescent protein which following Cre activity is excised, allowing a red fluorescent protein to be expressed[42]. During the imaging period, no cell death was observed in the control cells (Fig. 2A–C, M). In contrast, cell death was evident soon after excision of the Cre reporter in progenitors (driven by either CAG or Tbr2 promoters) (Fig. 2D–I, M, P–S, V). In these treatments, cell death occurred shortly (1–5 h) after Cre was active (Fig. 2 and Supplementary movies 1–4). After two days in culture, the percentage of viable cells expressing an excised Cre reporter in the progeny of both treatments was similar. A higher proportion of viable cells expressing excised Cre- reporter was seen when the Ta1 promoter was used (Fig. 2V). Despite attenuated death dynamics, postmitotic cells were sensitive to Hnrnpu loss of function. These results were recapitulated in organotypic slices obtained from Hnrnpu[fl/fl] co-electroporated with a Cre-ERT2 expressing plasmid and the CAG::SL Cre-reporter. Organotypic slices were treated with tamoxifen and subjected to time-lapse imaging. Whereas only a few red nuclei (indicating an excised Cre reporter) were observed when no tamoxifen was added, multiple red nuclei were noted following Cre activation by tamoxifen treatment (Supplementary Fig. 2B, D). Cre-expressing cells developed nuclear blebbing indicative of apoptosis less than 2 h after Cre-dependent excision of the reporter was noticeable

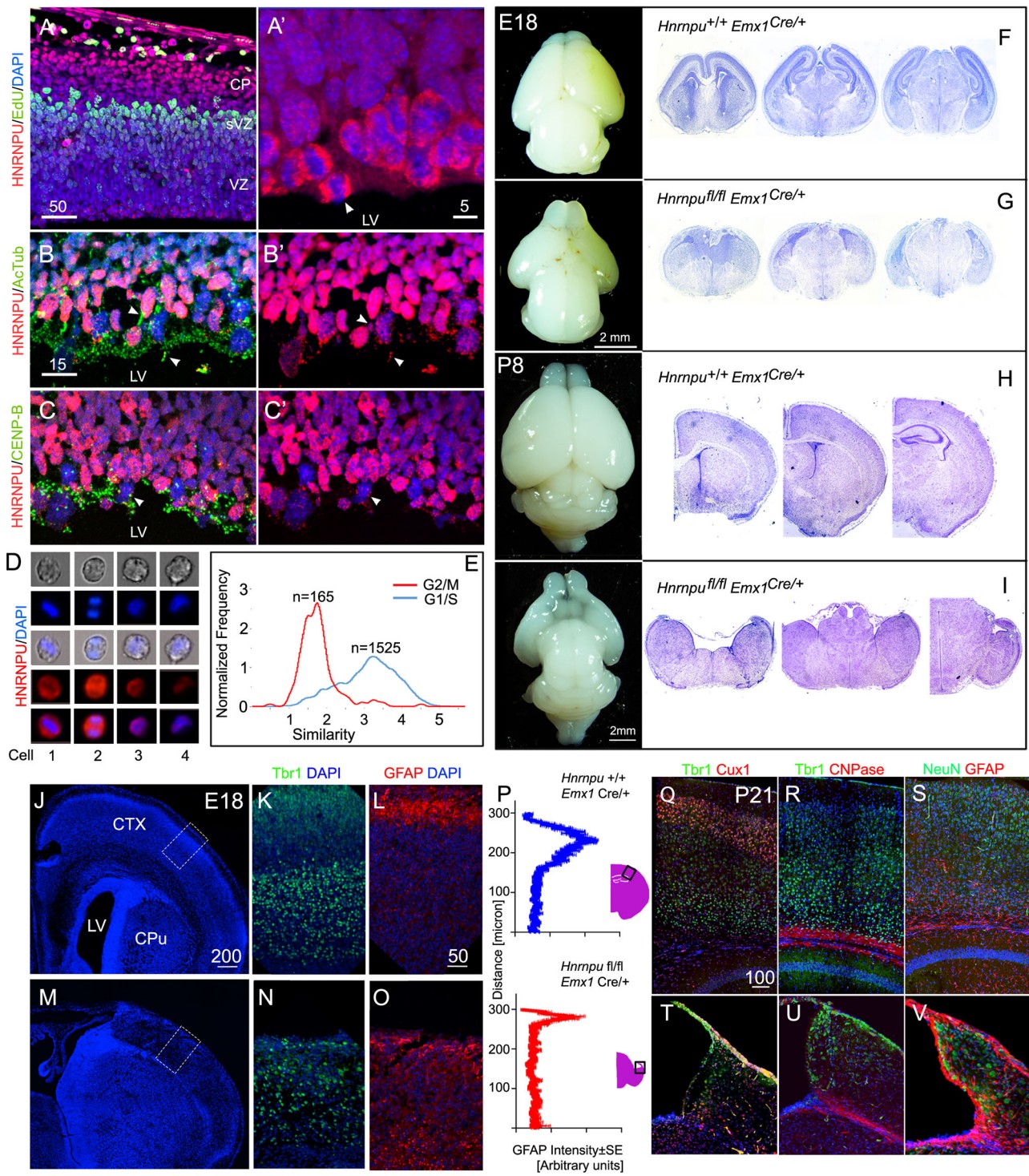

(Supplementary Fig. 2E). The tamoxifen-induced deletion was effective and visible by Western blot. Only the truncated HNRNPU protein (HNRNPU 1-276) was visible one day after tamoxifen induction of Cre in *Hnrnpu*fl/fl / *Ubc::Cre-ERT2* derived neurospheres (Supplementary Fig. 2F). Immunostaining of E14 cortices taken from control and mutant embryos with anti-SOX2 and TBR2 antibodies conferred the loss of proliferating cells in the neocortex of mutant animals consistent with the observed reduction in the cortex volume (Supplementary Fig. 3).

**HNRNPU affects gene expression and splicing**. We performed transcriptomic analyses to gain insights into the molecular

mechanisms underlying the observed pathology of *Hnrnpu* deletion. We generated RNA-seq libraries from cortices derived from E13 *Hnrnpu*fl/fl *Emx1*Cre/+ (Homo), *Hnrnpu*fl/+ *Emx1*Cre/+ (Het) embryos, and control littermates (WT) (Fig. 3). Our findings suggest that HNRNPU strongly affects gene expression. We detected 1556 differentially expressed (DE) genes between the Homo and the control cortices, and 346 DE genes were detected in control versus Het comparison (pAdj<0.05, log2FC > |1|, BM > 20, Fig. 3A and Supplementary Data 1). We noted a difference in the GO terms related to the upregulated and downregulated genes. The upregulated genes were related to synaptic activity, whereas the downregulated ones included several terms

**Fig. 1 HNRNPU is strongly expressed in mitotic mouse neural progenitors and is essential for brain development. A**-**C**′ Expression pattern of HNRNPU (Red) in coronal sections of E13 embryos cortices. **A** EdU, (Green), 1 h post-injection highlights cells in the S-phase at the basal border of the ventricular zone. **B** Acetylated tubulin (green) marks stabilized tubulin at the radial progenitor primary cilia. **C** CENP-B (green) marks centrosomes tethered to the apical surface. **D, E** Flow Cytometry of dissociated neurospheres (E13, 2 days in vitro) at four stages of the cell cycle, classified by EdU incorporation and DAPI stain. Red- HNRNPU, blue-DAPI. **E** Distribution of the levels of HNRNPU and co-localization with DAPI (Similarity) in a population of dissociated neurospheres classified as G2/M $n = 165$ or G1/S $n = 1525$. Images of whole brains and corresponding 5 μM thick coronal sections (Nissl staining) of embryonic day 18 (E18, **F, G**) and postnatal day 8 (P8), of $Emx1^{Cre}$ littermates carrying WT (**F, H**) or floxed $Hnrnpu$ alleles (**G, I**). **J–O** Expression of GFAP (Red) and Tbr1 (Green) in section of E18 obtained from control ($Hnrnpu^{+/+}$ $Emx1^{Cre/+}$) and mutant ($Hnrnpu^{fl/fl}$ $Emx1^{Cre/+}$) littermates. Insert indicate region magnified in **L, O**. Tbr1 images (**K, N**) were captured from equivalent locations. **P** Averaged GFAP intensity histograms in arbitrary units after background subtraction along the upper 300 μM of coronal sections of E18 cortices ($n = 3$, each measured in triplicates). **Q–V** P21 brain sections of mutant and control ($Hnrnpu^{+/+}$ $Emx1^{Cre/+}$) littermates stained with cortical plate layers markers (Tbr1, Cux1 in **Q, T**) and glia markers, 2′,3′-cyclic nucleotide-3′-phosphodiesterase (CNPase **R, U**) and GFAP (**S, V**). Schematic representation of the slice morphology (pink) showing images location (black rectangle). LV-lateral ventricle, CPu- caudate putamen, CTX- Cortex. Size bars are in μM.

associated with DNA (Supplementary Fig. 4A). The majority of the DE genes in the Homo had elevated expression. The DE genes contained, on average more exons. They were longer than the average length of the genome-wide gene (averages 7.4 exons vs. 5.4 and 3178 bp vs. 2293, Homo vs. control, respectively, highly significant by Wilcoxon test, $p < 2.2e$-16). Ingenuity pathway analysis of the DE genes identified multiple affected upstream regulators, including TP53, REST, and ASCL1 (Fig. 3B). Affected cellular activities included cell movement, migration, synaptogenesis, and synapse formation (Fig. 3B). Multiple signaling pathways, including synaptogenesis, neuroinflammation, reelin signaling, and cell cycle control, were also affected (Fig. 3B and Supplementary Fig. 4). The DE genes related to synaptogenesis and glutamate receptor signaling changed in Homo and Het brains, including calcium and potassium channel subunits, glutamate receptors, etc. Their differential expression was verified by qPCR (Fig. 3C).

Given the roles of HNRNPU in RNA splicing, we analyzed the RNA-seq data to identify differences in isoforms detected in the $Hnrnpu$ mutant mice compared with wildtype using MAJIQ[43]. We saw 850 differentially spliced genes comparing the homozygote mutants and the wildtype samples and 165 differentially spliced genes comparing the Het and the wildtype (Supplementary Data 2). When comparing the homozygote mutants versus the wildtype, 1187 local splicing variations (LSV) were detected, and 1054 (88%) involved exon skipping. In 331 of the LSV events, an alternative 3′ or 5′ splice site was detected. In only 117 LSV events, intron retention was detected. Accordingly, we detected 164 LSVs when comparing the heterozygotes to the wildtype, of which 140 LSV involved exon skipping (85%) and 54 LSVs with alternative 3′ or 5′ splice sites. Twenty-five LSVs events included intron retention. The list of genes was subjected to over-representation analysis[44] using a gene ontology database (Fig. 3D)[45,46]. The most enriched cellular components amongst the mis-spliced mRNAs included the synapse and the cytoskeleton (see DE gene analysis, Fig. 3B). We verified several alternatively spliced gene events ($Dcc$, $Siva1$, and $Mdm2$, Fig. 3E). The alternative splicing of $Dcc$ by NOVA has been implicated in neuronal migration and axonal guidance[47,48]. $Siva1$ is an apoptotic regulatory protein involved in synaptic function and is a downstream target of p53[49–51]. $Mdm2$, a critical negative regulator of the P53 protein, was identified as an alternatively spliced gene. Interestingly, the major MDM2 isoform expressed in $Hnrnpu^{fl/fl}$ $Emx1^{Cre/+}$ mutant cortices represented a higher abundance of the MDM2 isoform that lacked exon three. This exon encodes for a sequence required for TP53 binding, and its exclusion is less frequent in the wild type. The physical interaction and activity of the E3 ubiquitin ligase MDM2 is a central regulatory mechanism that keeps in check the low levels of TP53. Therefore, the absence of the binding domain facilitates TP53 activation[52–61].

**HNRNPU loss of function activates TP53 mediated cell death.** Since $Hnrnpu$'s mutation causes cell death, we examined whether culturing cortical embryonic neural progenitors as primary neurospheres can recapitulate this phenotype. We generated neurospheres from E13 mouse cortices, electroporated them with CRISPR/Cas9 $Hnrnpu$ gRNA plasmids, and conducted a single-cell RNA-seq experiment using the 10× Genomics© platform (Supplementary Fig. 5). We found that cells with low $Hnrnpu$ expression had increased TP53 pathway activity, whereas, within cells, with high $Hnrnpu$ levels in the same culture, this pathway was suppressed. To further validate this finding, we followed the dynamic stabilization of TP53 in cultured neurospheres subjected to the same treatment, as well as in mutant brain sections (Fig. 4A–F). In treated neurospheres, cell numbers decreased as early as 12 h after introducing $Hnrnpu$ sgRNA, coinciding with TP53 accumulation. Neither TP53 accumulation nor reduced cell numbers were evident in control cells (Fig. 4A, B). Accumulation of TP53 was detected in vivo in areas where the progressive loss of cortical structure was apparent (Fig. 4C–F). Using RNA derived from mutant and wildtype cortices, we verified the elevated expression of several P53 target genes $Arpp21$, $Perp$, $Gas6$, $Sat1$, $Dcxr$, $Anxa$, $Apoe$, $Atf3$, $Bbc3$, $P21$, and $Puma$ (Fig. 4G). These are a few genes among the 272 genes identified as TP53-regulated using Ingenuity analysis (Fig. 4G). Further, we confirmed the aberrant splicing of several $Tp53$ pathway genes (Fig. 4G, Supplementary Fig. 6). A schematic representation of the possible network connections between most verified genes related to the pathway is shown in Fig. 4H. Small molecule compounds were used to gain more insights regarding the death mechanisms driven by P53 accumulation. We followed their effect on neurospheres treated with $Hnrnpu$ sgRNA (Fig. 4I, I'). The pan-Caspase inhibitor, N-benzyloxycarbonyl-Val-Ala-Asp-fluoro-methyl ketone (Z-VAD-fmk), increased cell viability and proliferation indicated by neurosphere size and the normalized number of cells in the S-phase that incorporated EdU added to the media thirty minutes pre-fixation. The difluorophenoxymethylketone-based broad-spectrum Caspase inhibitor Q-VD-OPH was more effective than Z-VAD-fmk and has been previously reported to be less cytotoxic[62] (Fig. 4I, I'). Furthermore, treatments with either Nec-1 or pifithrin-mu (pFT-μ) promoted NPC viability suggesting the involvement of both canonical and non-canonical TP53 dependent death, including necroptosis[1,2].

**Genetic means to rescue $Hnrnpu$ deficiency.** Taking into consideration that TP53 is a direct trans-activator of numerous proapoptotic genes, we postulated that $Tp53$ deletion could rescue the observed cell death in the developing brains. To that end, we crossed $Hnrnpu^{fl/+}$ $Emx1^{Cre/+}$ $Tp53^{loxP/+}$ with $Hnrnpu^{fl/+}$ $Tp53^{loxP/+}$ (Fig. 5A). We observed that only when both alleles of $Tp53$ were deleted ($Hnrnpu^{fl/fl}$ $Emx1^{Cre/+}$ $Tp53^{loxP/loxP}$) the

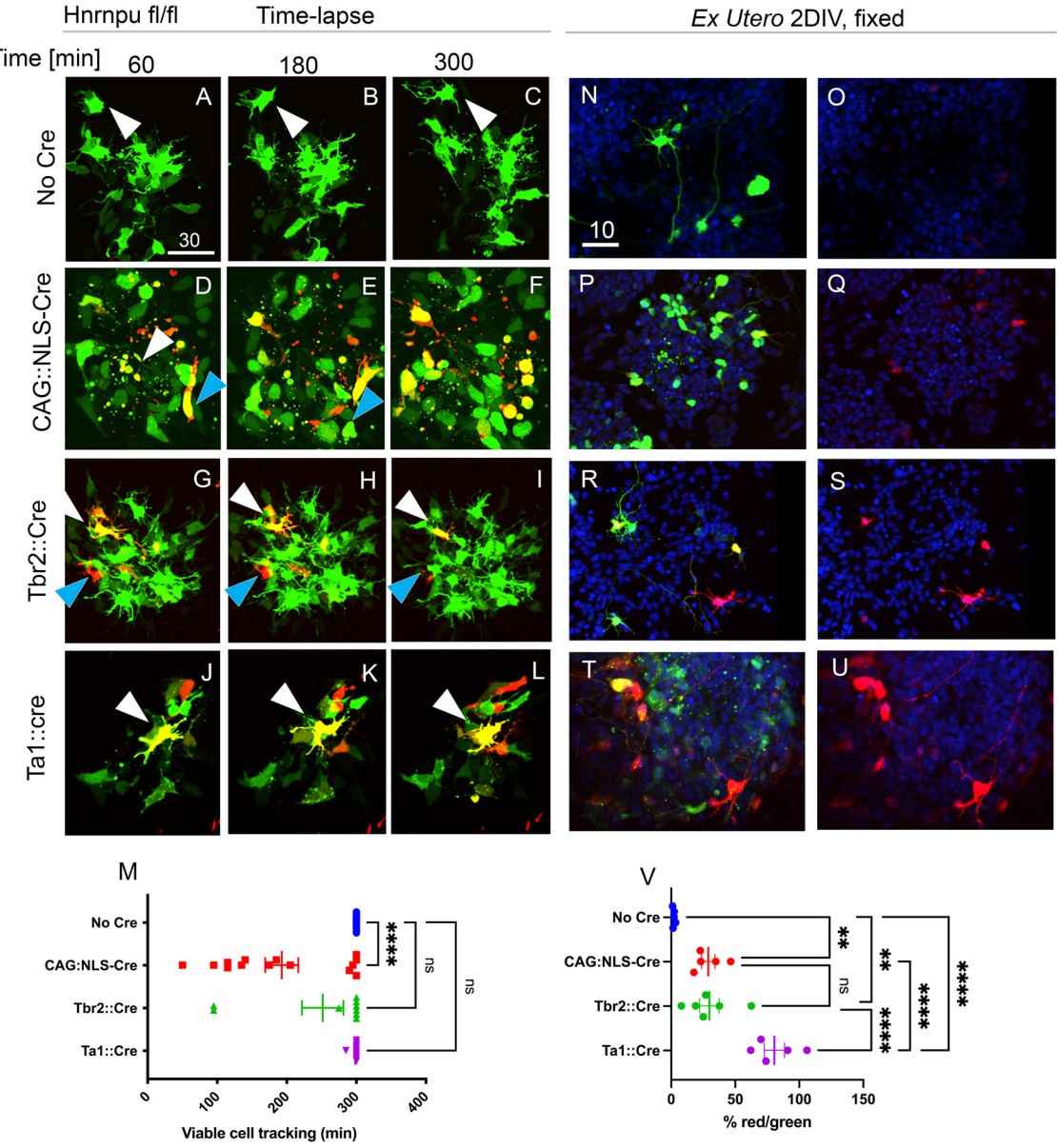

**Fig. 2 Neural progenitors are highly sensitive to HNRNPU loss.** Dual Color Cre reporter (CAG::SL) was electroporated to E13 *Hnrnpu* [fl/fl] embryos, injected with either no Cre (**A–C**, N-0, Supp_Movie 1) or with Cre expressing plasmids under one of three promoters (CAG::NLS-Cre-GFP (**D–F, P, Q**, Supp_Movie 2), Tbr2::Cre and (**G–I, R, S**, Supp_Movie 3) Ta1::Cre (**J–L, T–U**, Supp_Movie 4) **A–L** Images of time-lapse recording (5 h long) of primary cultures prepared from electroporated cortices. Arrowhead highlight specific cells in the field of view. **M** Tracking of cell viability in each field of view, a dot indicates the last time point a tracked cell is visible before disintegration, vertical lines indicate average values, bars indicate ±SEM range. Statistical analysis was performed using one-way ANOVA with Dunnett's multiple comparison tests, $p < 0.0001$. Control, $n = 17$, CAG::NLS-Cre-GFP $n = 14$, Tbr2::Cre, $n = 9$ and Ta1::Cre $n = 10$. **N–U** Images of fixed primary cultures, after 2 days in vitro (2DIV), Showing transfected cells (expressing floxed ZsGreen, Green) and cells with an excised reporter (mCherry, Red). **V** Percentage of cells expressing mCherry out of ZsGreen expressing cells in Control (**N, O**, $n = 581$) CAG::NLS-Cre (**P, Q**, $n = 787$), Tbr2::Cre (**R, S** $n = 182$) and Ta1::Cre (**T, U**, $n = 137$). Blue, DAPI. Statistical analysis was done using ordinary one-way ANOVA with uncorrected Fisher's LSD, p values: no Cre vs. CAG::NLS-Cre 0.0092, No cre vs. Tbr2::Cre $p = 0.0056$. No Cre vs. Ta1::Cre $p < 0.0001$. CAG::NLS-Cre vs. Tbr2::Cre, NS. CAG::NLS-Cre vs. Ta1::Cre $p < 0.0001$. Tbr2::Cre vs. Ta1::Cre $p < 0.0001$. **$P < 0.01$, ***$P < 0.001$, ****$P < 0.0001$, ns-non significant. Variability is indicated by ±SEM bars. Size bars units are µM. Source data are provided as a Source Data file.

immunoreactivity of Caspase 3 cleavage in the VZ was markedly reduced, and telencephalic derived structures were visible, indicating a clear reduction in cell death (Fig. 5B–D). The deletion of *Tp53* did not affect the splicing of *Mdm2* (Supplementary Fig. 7A). Some of the rescued cells were cycling cells. The number of cells that incorporated EdU within 30 min or that were labeled by anti-phospho-Histone H3 antibodies, or KI67 antibodies, increased. However, their distribution was usually ectopic, with proliferating cells appearing outside the VZ/sVZ (Fig. 5E–G,

Y-AA). The number of cells incorporating EdU (30 min pulse) in the mutant brain was reduced to 31.7% ± 3.07 of control values (Average ±SE, of $n = 3$ brain). These numbers were partially restored to 56.97% ± 2.65 compared to the control in the rescued brains (Fig. 5AE). It was possible to visualize an increase in the number of intermediate progenitors (TBR2[+]) that appeared in the expected location in the rescued sVZ (Fig. 5V–X). To better understand the transcriptional profile of the rescue phenotype, we conducted 3'-RNA-seq (MARSeq) from the cortices of three

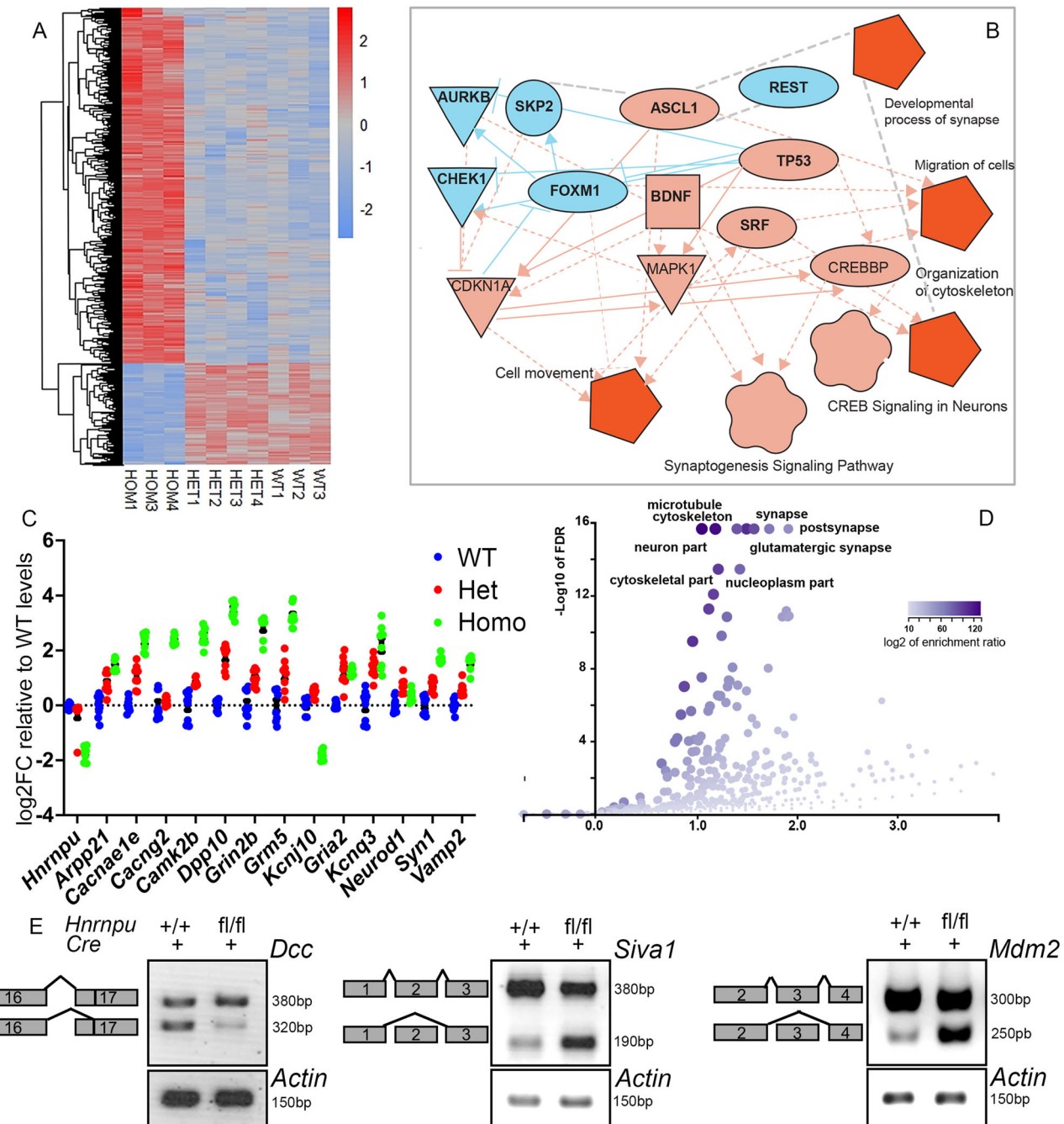

**Fig. 3 *Hnrnpu* mutant cortices display changes in mRNA expression and alternative splicing of genes critical for brain development and function.**
**A** Hierarchical clustering of the expression of 1556 cortical DE genes in E13 control (*Hnrnpu*$^{+/+}$*Emx1*$^{Cre}$), Mutant (*Hnrnpu*$^{fl/fl}$ *Emx1*$^{Cre/+}$) and Heterozygous (Het, *Hnrnpu*$^{fl/+}$ *Emx1*$^{Cre/+}$) E13 mouse cortices. The expression values are represented by the log-normalized read count after the Zscore transformation. Each row is a gene. **B** Ingenuity analysis of RNA-seq highlighting multiple affected pathways and upstream regulators (Shape code: pentagon- function, flower- canonical pathway, ellipsoid- upstream regulator, triangle- kinase, square- growth factor, circle- other. Color code: light blue- downregulation, Red- strong activation, Pink- activation. lines represent interconnections between components in the network. color code: light blue- downregulation, Red- strong activation, Pink- activation). **C** Verification of differential expression of selected genes by qPCR in E13 cortices of control, Mutant, and Het embryos (*n* = 3 biological repeats (each measured in triplicates). Error bars ± SEM. **D** Dysregulation of alternative splicing landscape in *Hnrnpu* mutant mice identified by MAJIQ. The volcano plot depicts the comparison between the mutant (*Hnrnpu*$^{fl/fl}$*Emx1*$^{Cre}$) and the wildtype (*Hnrnpu*$^{+/+}$*Emx1*$^{Cre}$). Enriched cellular components are indicated. **E** Splice variant-specific PCR primers detect alternative isoforms of DCC, Siva1, and MDM2, in cDNA prepared from E13 embryo cortices of mutant (*Hnrnpu*$^{fl/fl}$*Emx1*$^{Cre}$) and control (*Hnrnpu*$^{+/+}$*Emx1*$^{Cre}$) littermates. Source data are provided as a Source Data file.

groups representing mutant, control, and rescue (Fig. 5A, H and Supplementary Data 3). Whereas the number of DE genes in control versus mutant comparison was 640, in the rescue versus mutant comparison, a mild increase in the number of DE genes

was observed (737). The DE genes detected in control versus mutant comparison were subjected to GO-term analysis. The same terms found in the RNA-seq mentioned previously were detected. Only 148 DE genes were detected in the control versus

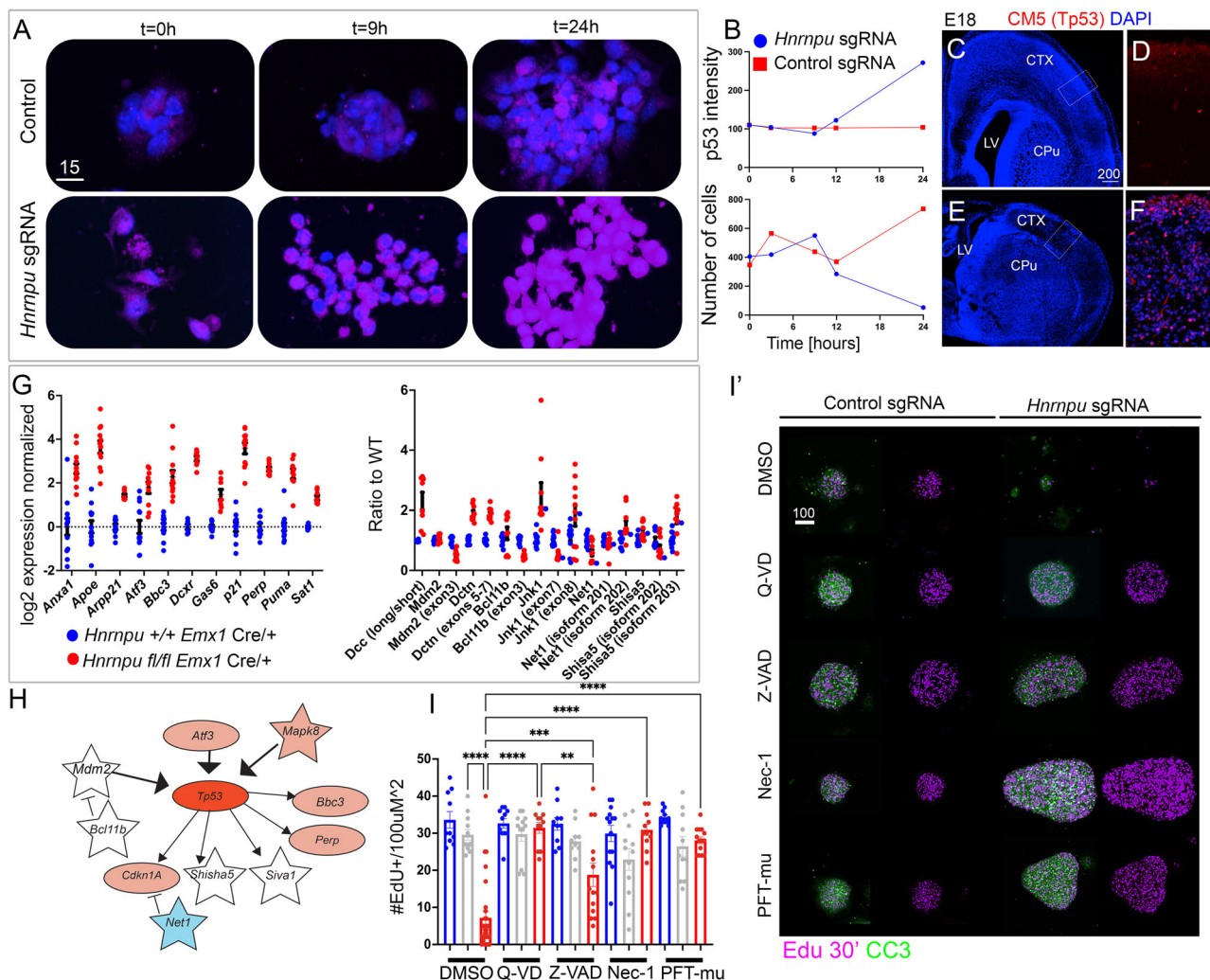

**Fig. 4 Activation of TP53-mediated apoptotic pathway following *Hnrnpu* KO. A** Time-dependent accumulation of TP53 (CM5 antibody) in neurospheres treated with control CRISPR/CAS9 plasmid (px330) or *Hnrnpu* sgRNA's and plated at indicated time. Acquisition of the intensity of the CM5 signal was done in identical imaging parameters for all images. **B** Measurements of the dynamic accumulation of TP53. Cells were identified DAPI, and the average intensity in each spot was normalized to a non-specific background (upper panel). The accumulative number of cells in all fields of view is plotted against time (lower panel). **C**–**F** Low magnification images (Dapi, blue) of coronal sections (E18) of control (*Hnrnpu* $^{+/+}$*Emx1*$^{Cre/+}$) and mutant (*Hnrnpu*$^{fl/fl}$*Emx1*$^{Cre/+}$) brains. Inserts indicate the location of high magnification images (**D**, **F**). Anti-TP53 (CM5, Red) fails to react with sections from control cortices. **D** Reveals stabilization of TP53 in cells throughout mutant cortices (**F**). **G** Normalized expression levels (left panel) and alternative splicing (right panel) in the mutant (*Hnrnpu*$^{fl/fl}$*Emx1*$^{Cre}$) and control (*Hnrnpu*$^{+/+}$*Emx1*$^{Cre}$) E13 cortices, showing elevated expression of *Tp53* targets and misrepresentation of alternative splice variants of genes in the TP53 pathway $n = 3$ biological repeats (each measured in triplicates). Error bars ±SEM. **H** Schematic representation of functional connections between differentially expressed (ellipsoid) and abnormal spliced (star) gene products presented in **C**. color code: Red-Strong activation, Pink-overexpression, Blue- downregulation, White- no change in expression levels. $n = 4$ biological repeats **I-I′** Effective protection from apoptotic cell death of CRISPR/CAS9 sgRNA (*Hnrnpu* sgRNA, Red) Non-electroporated neurospheres (gray) and control (px330, blue) Electroporated neurospheres were treated with solvent only (DMSO), Q-VD-OPh (50 µM), Z-VAD-fmk (50 µM), Necrostatin-1 (Nec1, 1 µM) and Pifithrin-µ (pFT-mu, 5 µM). Bars indicate the number of EdU+ cells per 100 µM$^2$ of the neurospheres surface area, following 30 min exposure to EdU. Ordinary one-way ANOVA with Tukey's multiple comparison test was used for data analysis. P values: **$p < 0.01$, ***$p < 0.001$, ****$p < 0.0001$. Error bars (±SEM) are indicated **F** Images of neurospheres treated with either DMSO or indicated small molecules. EdU incorporation (Click chemistry, magenta) and Cleaved Caspase, Asp175 (CC3, Green), are presented. Size bars units are µM. Source data are provided as a Source Data file.

rescue comparison, suggesting that the expression of many genes was restored. Notably, the expression of p53 responsive genes was restored in the rescue cortices.

We observed that the expression of genes in different pathways was more similar to control levels and not only those related to the *Tp53* pathway. This is due to the high level of crosstalk exhibited in the network of pathways. Pathway analysis revealed that the rescue differed from the control in several signaling pathways, including BDNF, synaptogenesis, CDK5, and cytoskeletal organization (Fig. 5I). Non-rescued genes with a differential expression included

those involved in RNA splicing (e.g., *Hnrnpu*, *Hnrnpd*, *Rbfox1*, and *Rbfox3*). Unlike the mutants, the rescued embryos developed a cortical plate with the laminar organization of TBR1 and CTIP2 positive cells, yet its width was not fully restored (Fig. 5J–R). The *Reln* mRNA and its protein product were upregulated in the mutants. The dramatic increase in protein levels was shown in mutated cortices (Fig. 5J, K). This phenotype was corrected in the rescued embryos (Fig. 5L). We observed a restoration of the apical F-actin belt, as well as a significant improvement in the overall actin organization following *Tp53* loss (Fig. 5AB–AD).

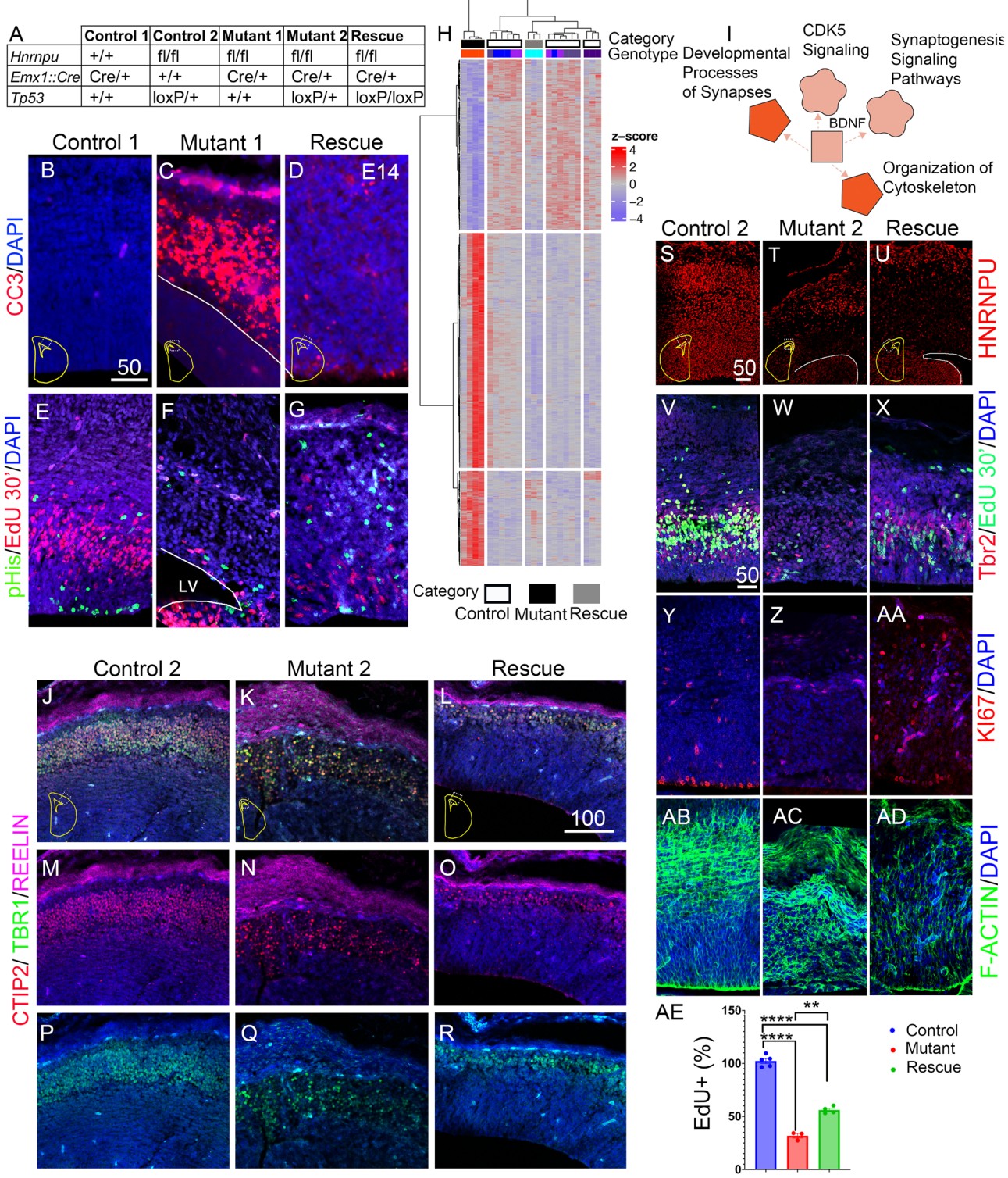

| | Control 1 | Control 2 | Mutant 1 | Mutant 2 | Rescue |
|---|---|---|---|---|---|
| *Hnrnpu* | +/+ | fl/fl | fl/fl | fl/fl | fl/fl |
| *Emx1::Cre* | Cre/+ | +/+ | Cre/+ | Cre/+ | Cre/+ |
| *Tp53* | +/+ | loxP/+ | +/+ | loxP/+ | loxP/loxP |

Nevertheless, the rescue was partial. The F-Actin alignment in the apical aspect of the VZ was partially reestablished, but the overall actin levels were lower than those in control littermate cortices (Fig. 5AB–AD). The organization of the centrosomes of the radial glia cells at the apical surface was disrupted in the mutant brains and was significantly improved in the rescue sections (Supplementary Fig. 7B–G). In our mouse model, *Hnrnpu* is mutated in *Emx1* positive cells only. To verify that the viable cells in the cortex are deleted, we immunostained the sections with antibodies that recognize the N-terminus of the

truncated HNRNPU protein (Fig. 5S–U). The majority of the cortical-plate cells in the sections of the rescued embryos exhibited reduced expression of HNRNPU similar to those seen in the mutant brains, suggesting that the truncated protein is unstable.

We next examined genetic means to lessen the observed cell death in neurosphere cultures. In addition to manipulating the levels of P53 either by CRISPR-mediated deletion of *Tp53* or by ectopic expression of exon 3-containing *Mdm2*, we also tested CRISPR-mediated deletion of *Srsf3* (Fig. 6A–D). SR (serine/

**Fig. 5 Genetic ablation of *Tp53* reverses cortical loss in *Hnrnpu* mutant embryos. A** The classification of the three alleles. Accumulation of CC3, Red in control 1 (**B**) and Mutant 1 (**C**). Reduced levels of p53 (CM5, Red) in *Hnrnpu*^fl/fl^ Emx1^Cre/+^, with *Tp53*-/- (**D**). **E** pHis, Green, combined with EdU (30 min post-injection, magenta). **F** Mutant brains with reduced signals. EdU and pHis staining in mutant brains with deleted Tp53 (**G**). **H** Hierarchical clustering of 581 genes which show differential gene expression. Heatmap of 581 annotated genes showing differential expression between three categories (Control-white, Mutant-Black, and Rescue-Gray. The Mutant samples (Red) are all homozygous for the floxed *Hnrnpu* allele. The rescue category (gray) includes a single genotype, *Hnrnpu*^fl/fl^ Tp53^loxP/loxP^ (light blue). The "Control" category (white) includes four different genotypes *Hnrnpu*^fl/+^ Tp53^loxP/loxP^ (Blue) *Hnrnpu*^fl/+^ Tp53^loxP/+^ (Purple), *Hnrnpu*^+/+^ Tp53^loxP/loxP^(Gray), and *Hnrnpu*^+/+^Tp53^loxP/+^ (Magenta). **I** Ingenuity pathway analysis of 148 DE genes in rescued brains vs. control (Red- Strong activation, Pink- activation) **J–R** Coronal sections of E14 cortices from Control, *Hnrnpu* Mutant 2 carrying a single Tp53 deleted allele and Hnrnpu/Tp53 double mutant, stained for Reelin (Magenta), Tbr1 (Green) and CTIP2 (Red). Single Tp53 allele deletion (Mutant 2), stained with Rabbit N-ter Anti-hnRNP U antibody (Red) in the Control brain section (**S**) and Mutant 2 (**T**) and Rescue (**U**) samples. Tbr2 + (Red) in the VZ and sVZ (**V**), EdU + (Green), mutant cortices (**W**) and rescued brains (**X**). KI67 + in Control (**Y**), Mutant 2 (**Z**), and Rescued embryos (**AA**). Phalloidin 488 (Green) stains F-Actin in Control (**AB**) and mutant sections (Mutant 2, **AC**). F-actin belt (Phalloidin 488, Green) (Control Cortex, **AB**) in the mutant brain (Mutant 2, **AC**) and rescued cortices (**AD**). **AE** Proportion of EdU positive cells ±SEM (30 min pulse) counted in VZ/sVZ of control (blue), mutant (red) and rescued (green) cortices (n = 3). Nested *t* test. P values Control vs. Mutant *p* < 0.0001, Control vs. Rescue *p* < 0.0001, Mutant vs. Rescue *p* = 0.0067. \*\**p* < 0.01, \*\*\**p* < 0.001, \*\*\*\**p* < 0.0001. LV-lateral ventricle⊡ Size bar units are µM. Source data are provided as a Source Data file.

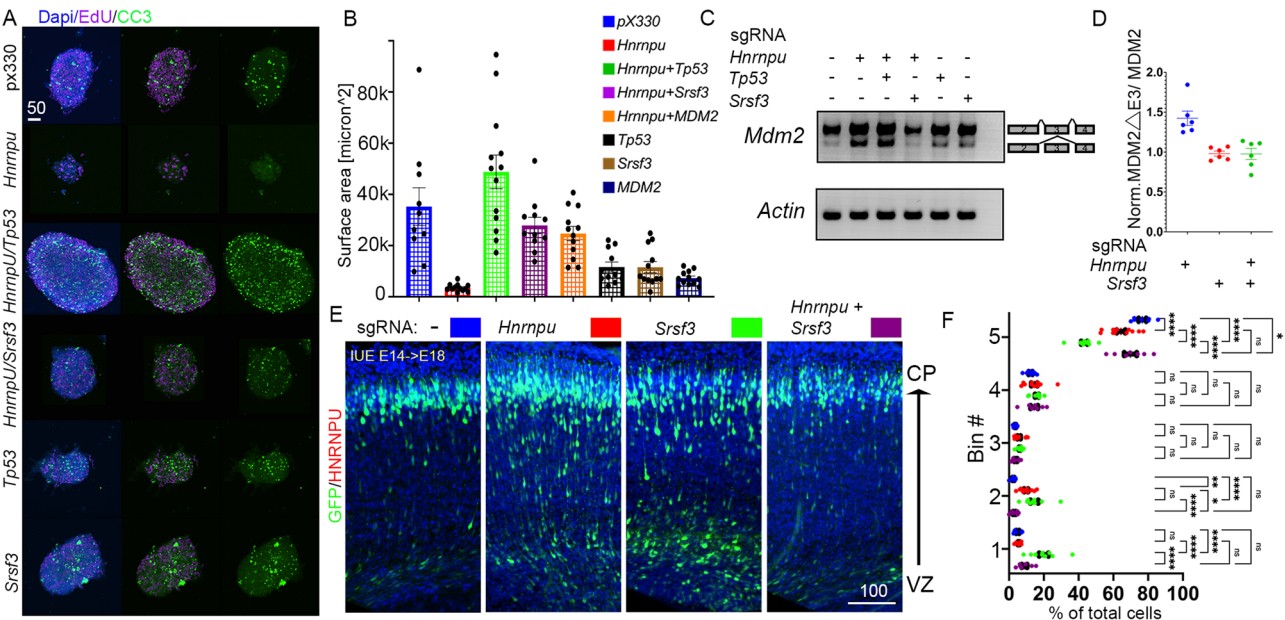

**Fig. 6 *Srsf3* downregulation compensates for HNRNPU loss of activity. A** Images of E14 derived mouse cortex neurospheres electroporated with either px330 (CRISPR/CAS9 Control plasmid) or with two sgRNA sequences, for either *Hnrnpu*, *Tp53*, or *Srsf3*. When genes are targeted (*Hnrnpu* and *Tp53* or *Hrnrnpu* and *Srsf3*), neurospheres grow to large sizes (DAPI, Blue) and incorporate EdU (Magenta) despite the accumulation of activated Caspase 3 (CC3, green). **B** Quantification of the size of neurospheres, two days post electroporation treated with the indicated combination of CRISPR/CAS9 sgRNA's or overexpressing MDM2. Px330, n = 10 *Hnrnpu* sgRNA n = 12 *Hnrnpu* + *Tp53* sgRNA n = 13 *Hnrnpu* + srsf3 sgRNA n = 11 *Hnrnpu* sgRNA +MDM2 n = 12 Tp53 sgRNA n = 11 *Srsf3* sgRNA n = 12 MDM2 n = 12. Error bars ±SE. Analysis was done using ordinary one-way ANOVA with Dunnett's multiple comparisons test. **C** Splice variants sensitive PCR of cDNA prepared from treated neurospheres (depicted in panels A and B) shows correction of the over-representation of MDM2 Exon 3 skipping splice variant (300 bp vs. 250 pb, actin band size: 150 bp). One-way ANOVA with Dunnett's multiple comparison test. Adjusted *p* values vs. control: *Hnrnpu* < 0.0001, *Hnrnpu*+*Tp53*, *Hnrnpu*+ *Srsf3* NS, *Hnrrpu* + *MDM2* NS, *Tp53 p* = 0.0007, *Srsf3 p* = 0.0005, *MDM2 p* < 0.0001. Error bars (±SEM). **D** qPCR of the relative abundance of MDM2 isoforms in cultured Neurospheres (NS). Graphs depicted transcript levels, compared to control (p x 330, no sgRNA) of MDM2 splice variants (MDM2 lacking exon 3 (MDM2ΔE3)/MDM2 with retained E3 (MDM2) in NS treated with the indicated combination of CRISPR/CAS9 sgRNA's, n = 6. Error bars (±SEM) are indicated. Coronal sections (60 µM) shows the location of radially migrating cortical neurons electroporated with the control plasmid (blue), *Hnrnpu* sgRNA (red), *Srsf3* sgRNA (green), both genes (purple). **E, F** Distribution GFP + cells (%) in five horizontal bins across the cortex width (1 the most apical), Averages, and error bars (±SEM). VZ-Ventricular Zone, CP-cortical plate. Three biological repeats were used; 2–4 70 µM thick slices of comparable position were analyzed of each electroporated brain. Control n = 8, *Hnrnpu* sgRNA n = 11, *Srsf3* sgRNA n = 9, *Srsf3* + *Hnrnpu* sgRNA n = 7. Analysis was done using two-way ANOVA with Tukey's multiple comparisons test. *P* values: \*\**p* < 0.01, \*\*\**p* < 0.001, \*\*\*\**p* < 0.0001, ns non significant. Source data are provided as a Source Data file.

arginine-rich) splicing factors (SRSF) generally oppose HNRNPs with exon skipping and exon retention[63,64]. Among the different SRSF proteins, we targeted SRSF3 because i) high levels of *Srsf3* expression were detected in our RNA-seq data; ii) depletion of SRSF3 promoted the inclusion of *SMN2* exon 7[65], where the exclusion of this exon is dependent on HNRNPU[33]. Indeed, *Srsf3* deletion was an effective means to circumvent Hnrnpu CRISPR-

mediated deletion outcomes. The reduced SRSF3 protein level in the absence of HNRNPU restored the ratio of *Mdm2* exon 3 inclusion to wildtype levels and reverted the size of the neurospheres (Fig. 6A–D). Further, *Hnrnpu*'s deletion affected genes implicated in cell migration and movement as well as the cytoskeleton, all of which are crucial for proper neuronal migration (Fig. 3B). We tested the effect of *Hnrnpu* deletion by

in utero electroporation on neuronal migration. *In utero* delivery of *Hnrnpu* sgRNA effectively reduced HNRNPU protein levels as estimated by staining plated cortical neurons. A small yet significant reduction of Srsf3 was also noted (Supplementary Fig. 8). Four days post electroporation, the treated cells were primarily negative for Cleaved Caspase 3 (Supplementary Fig. 9). The radial migration of pyramidal neurons born at E14 to layers II–IV was mildly impaired following CRISPR/CAS9 sgRNA targeting *Hnrnpu* and more severely affected when *Srsf3* was targeted (Fig. 6E, F). In utero delivery of a combination of sgRNAs targeting both *Srsf3* and *Hnrnpu* resulted in a comparable reduction of both proteins (Supplementary Fig. 10). The combined deletion of both RNA-binding proteins completely rescued the migration to the cortical plate in four out of five bins (Fig. 6E, F). Collectively, our data indicate that modulation of RNA splicing can reverse both impaired cell viability and neuronal migration resulting from HNRNPU loss of function.

## Discussion

*Hnrnpu's* truncation in the developing telencephalon using *Emx1*-driven Cre recombinase resulted in protein loss, massive cell death, and elimination of the cerebral cortex at an early postnatal stage. Neural progenitors and postmitotic neurons were vulnerable to loss of function of the HNRNPU protein. Nevertheless, our data do not rule out the possibility that a broader range of cell types contributes to the dramatic cortical loss following Cre activation in Emx1 expressing cells. Based on the different cell death dynamics that we observed in vitro using CAG, Tbr2, or Ta1 promoter-driven HNRNPU truncation, we postulate cell-specific sensitivity to HNRNPU loss of function may exist. This could partly reflect the relatively high expression of several anti-apoptotic genes in postmitotic neurons. *Bcl2l1, Mcl1*, and *Bcl2l12* were highly expressed in the RNA-seq data. The consequences of crossing the same mouse model with other Cre recombinase lines also demonstrated cell-type-specific sensitivity. A constitutive deletion using a germ-line-expressing *Sox2*-Cre line did not result in viable offspring[34], confirming previous findings[14]. The genomic deletion of *Hnrnpu* in cardiomyocytes and skeletal muscle using muscle creatine kinase-Cre (*Ckmm*-Cre) resulted in lethal dilated cardiomyopathy at P14[34]. In the same study, mouse lethality at P10 was observed using a cardiomyocyte-specific Cre line, cardiac myosin heavy chain-α Cre (*Myh6*-Cre)[34]. Compared to our findings, apoptosis was only slightly increased in the *Hnrnpu* mutant hearts. Gene expression analysis from P14 hearts detected 734 DE genes, and similar to our results, more genes were upregulated (438) than downregulated (296). Comparison analysis of both data sets detected the highest scoring gene signature, "Organismal death". Several common DE genes included ephrinB3, and its receptor EphB3, which have been implicated in cell survival, axonal pathfinding, and neuronal migration and were also involved in heart development[66–75]. Other common DE genes included subunits of glutamate receptors, channels, and transporters, which may contribute to neuronal hyperexcitability.

The expression of several genes was abnormal in our heterozygous mice, consistent with the epilepsy phenotypes detected in patients. The common DE genes in these two mouse models included a signature of the *Tp53* pathway activation and negative regulation of G1/S transition. Using the human skeletal actin promoter (HSA-Cre), a skeletal muscle-specific deletion did not affect the development and early postnatal growth. Still, it resulted in severe adult-onset myopathy[76]. Changes in gene expression were also noted in this study.

Our mouse model expressed reduced levels of a truncated protein, which was hardly detectable in mutant and rescued brains, suggesting protein instability. This truncated protein was designed to include chromatin/DNA binding domains (through the N-terminal SAP domain) but lacked the RNA binding motifs[77]. Notably, RNA-binding proteins can also regulate gene expression[64,78]. Although HNRNPU's role in splicing is well known, its targets in the developing brain are poorly defined[33–35,76,79]. The role of HNRNPU in RNA splicing depends on the spatial and temporal context. We found that most LSV involved skipped exons (88%), similar to the adult muscle (74%)[76]. However, in the developing heart, the fraction of retained introns was higher (40%) than that of skipped exons (35%)[34]. Pathway analysis of the differentially spliced genes indicated the involvement of HNRNPU in regulating the cytoskeleton and synapse formation. One alternatively spliced gene is *Dynactin-1*, which co-regulates the activity of the molecular motor cytoplasmic dynein with LIS1 and is critical in neuronal migration[80–82]. The alternatively spliced exon regulates the interaction between Dynactin-1 and microtubules[83].

We then focused on the role of TP53 in the death of cultured progenitors and on the complete loss of cortical structures in vivo; both events of transcriptional regulation and alternative splicing resulted in further activation of the pathway. Applying different cell-death inhibitors (e.g., Caspase, TP53, necroptosis inhibitors) rescued neurospheres' growth. In vivo, we inactivated the TP53-dependent cell death by crossing the mice with a floxed allele of *Tp53*. This genetic manipulation enabled cortical formation, increased progenitors proliferating, and partially restored the expression of layer markers. Overall, the cortices displayed mended organization and considerably improved centrosomal localization. Gene expression revealed a higher similarity between the rescued mice and the control ones, although the rescued cortices were abnormal. Ectopic positions of mitotic cells, actin organization, and cortical plate width were not restored to normal parameters. Of interest, it is possible to note that mutations in *Tp73*, a member of the TP53 protein family, affect airway motile cilia and cause lissencephaly[84].

HNRNPU is a centrosomal protein, and apoptosis has also been observed in case of mutations in other centrosomal proteins, yet not to the extent of the loss of the telencephalon[85–88]. Cell death occurred after conditional *Sas4* knockout or either *Cep63* or *Cpap* knockout[85–87]. When these mice were crossed with *Tp53*-/-, microcephaly was eliminated, yet, the mitotic cells were displaced. Crossing *Tp53*-/- with mice overexpressing PLK4 resulted in a partial rescue of brain size[88]. *Hnrnpu's* deletion in mitotic cells caused changes in upstream regulators of TP53 leading to the pathway's activation. However, the partial rescue observed suggests that in addition to TP53, other death pathways are activated.

We then considered ameliorating the observed phenotype by additional genetic means. In neurospheres, the co-deletion of *Tp53* and *Hnrnpu* significantly increased their size. Similarly, introducing an *MDM2*-expressing plasmid containing exon 3 significantly increased the size of the neurospheres, yet not to control levels. Impressively, the co-deletion of *Srsf3* and *Hnrnpu* resulted in neurospheres that were similar to controls. This treatment also partially corrected the splicing defect of *Mdm2*, reversing the ratio between the MDM2 splice variants to nearly control levels, namely, the higher relative abundance of the inclusion of E3 and possibly more efficient ubiquitination of P53. Using the same approach, we were able to reverse neuronal migration delays caused by HNRNPU loss of function. The expression and splicing of multiple genes are likely affected following the reduction of HNRNPU and SRSF3. At present, we do not have a comprehensive molecular analysis to explain the observed rescue phenotypes fully. Collectively, our data suggest that HNRNPU regulates splicing, which is critical for brain development.

## Methods

**Ethical regulations**. All animal work included in the manuscript is covered by accepted IACUC protocols. Animal protocols were approved by the Weizmann Institute Institutional Animal Care and Use Committee.

**Animal studies**. The mice were kept in the animal facility at a temperature of 22 °C ± 1 °C, 50% ± 10% humidity, and a 12 h light/dark cycle.

Conditional *Hnrnpu* allele received from Prof. Maniatis[34].

Emx1-Cre B6.129S2-Emx1tm1(Cre)Krj/J Stock No: 005628, The Jackson Laboratory.

*Tp53* $^{LoxP/LoxP}$ (Marino et al.,[89]). http://genesdev.cshlp.org/content/14/8/994.long.

UBC-Cre-ERT2 B6.Cg-*Ndor1*$^{Tg(UBC-Cre/ERT2)1Ejb}$/1J Stock No: 007001, The Jackson Laboratory Hsd:ICR(CD1) ENVIGO (Harlan). The study used both sexes; ages are indicated in each figure legend.

*Software and algorithms*. Imaris software 9.5.1 (Bitplane, Zurich, Switzerland) (RRID:SCR_007370)

GraphPad Prism 9, GraphPad Software, La Jolla California USA, www.graphpad.com.

Statistics and reproducibility: All Immunohistochemistry experiments were conducted on at least three different animals of each genotype that were collected throughout the study. In utero electroporation for migration and lineage tracing studies were done on multiple pregnant females, and a minimum of three brains were used for statistical analysis. Variant sensitive PCR experiments were performed in three biological repeats and two technical repeats. Gels were processed in parallel.

Neurospheres (NS) preparation and electroporation: Cortices of E13 or E14 embryos were dissected in oxygenated dissection media (cold L15 supplemented with Glucose (0.6%)). After meninges removal, the cortices were mechanically dissociated and immediately plated in low adherence plates in NeuroBasal media containing B27 (×1) GlutaMAX (×1) Gentamicin (25 µg/ml), EGF (20 ng/ml), bFGF (20 ng/ml), and Heparin (2 µg/ml). After 2 days, NS were dissociated with Accutase, washed with OptiMEM, and mixed with a DNA solution (4 µg of each pX330 CRISPR/CAS9 plasmid, 1 µg of CAG::GFP or CAG::dsRed, 4 µg of MDM2 cDNA, as indicated in each experiment). Dissociated cells (1–2 × 10$^6$) were electroporated in 2 mm cuvettes in Nepa21 Electroporator (poring parameters: 2 × 150 V, 3 ms long pulses, with a 50 ms interval and 10% decay rate. Transfer: 5 × 20 V pulses, 50 ms long, with 50 ms intervals and 40% decay rate. To obtain adherent NS, a growth factor depleted media (otherwise identical to that described) was used to plate the NS on PLL/Laminin coated cover glass for 1 h. For scRNA seq, neurospheres were prepared from E13 ICR cortices cultured for 2 days in the presence of EGF (20 ng/ml), bFGF (20 ng/ml), and Heparin (2µg/ml). Neurospheres were dissociated using Accutase and electroporated with a DNA mixture containing 4 µg of pX330 CRISPR/CAS9 Hnrnpu Guide A, 4 µg of pX330 CRISPR/CAS9 Guide B, and 1 µg of CAG::GFP. Cells were collected 24 h post electroporation, dissociated using Accutase, and passed through a 40-micron strainer. As estimated by cell count, 40% of the dissociated cells were GFP positive, and 90% were viable.

Imaging flow cytometry: Cortical neurospheres were prepared from E14 mouse embryos and were cultured for 2 days in vitro. Edu (10 µM) was added to the media 30 min before dissociation and fixation of the cells.

Fixed (4% PFA) and permeabilized cells were labeled with Edu-Click 488 flow cytometry kit (Alexa Flour 488 azide) and later stained with Rabbit N-ter anti-hnRNP U antibody, Abcam, and anti-Rabbit Alexa 647 and DAPI. Cells were imaged using a multispectral IFC (ImageStreamX mark II imaging flow-cytometer: Amnis Corp, Seattle, WA, part of Luminex, Tx). Lasers used were 405 nm (70 mW), 488 nm (40 mW), 561 nm (200 mW), 642 nm (20 mW) and for side scatter 785 nm (5mW), and the lens used was 40×. Channels used were Ch01 (Bright-field), Ch02 (EDU), Ch06 (Side scatter), Ch07 (DAPI), Ch09 (Bright-filed of camera 2) and Ch11 (hnRNPU). At least 5 × 10$^4$ cells were collected from each sample, and data were analyzed using the manufacturer's image analysis software (IDEAS 6.2; Amnis Corp). Cells were gated for single cells, using the area (the number of microns squared in a mask) and aspect ratio (the Minor Axis divided by the Major Axis) features on the bright-field channel. Focused cells were selected using the Gradient RMS feature and contrast features (measures the sharpness quality of an image by detecting large changes of pixel values in the image) on the bright-field image. Cropped cells were further eliminated by plotting the cell area of the bright field image against the Centroid X feature (the number of pixels in the horizontal axis from the left corner of the image to the center of the cell mask). The cell cycle was determined using the intensity (the sum of the background-subtracted pixel values within the masked area of the image) of the DAPI staining (Channel 7). Cells in the S-phase were further verified by plotting the EdU intensity (Channel 2) vs. DAPI intensity. Mitotic cells were identified by plotting the Bright Detail Intensity (the intensity of localized bright spots that are 3 pixels in radius or less within the masked area in the image, with the local background around the spots removed) against the area of the top 50% highest intensity staining (using the

Threshold_50 mask), of the DAPI channel. As the DNA is more condensed, mitotic cells could be identified by high bright detail intensity and low Area Threshold values. The co-localization of HNRNPU with the nuclear image (DAPI) was calculated using the Similarity feature (log-transformed Pearson's Correlation Coefficient between the two images). Values above 1.5 indicate co-localization.

Ex utero primary cultures time-lapse imaging: E13 embryos were removed from the uterus to a cold dissection media. Each embryo was injected with a DNA mix containing Cre expressing plasmid (1 µg) and a dual-color reporter (CAG::Stoplight, 1 µg). Electroporation parameters were identical to those used for in utero electroporation, described below, except for the former done bilaterally. Immediately after electroporation, the brains were mechanically dissociated and plated on PLL/laminin-coated glass. Time-lapse imaging was done 24 h post-plating, for 5 h, on a Leica DMi8 with an Andor Dragonfly 202, spinning disk confocal.

Construction of bulk MARS-seq libraries: Mouse embryo cortices (E13) were carefully dissected snap-frozen in liquid nitrogen until RNA was simultaneously extracted from 3 to 4 embryos of each genotype. Bulk MARS-seq libraries were produced from 50 ng of total RNA as previously described[90]. Libraries were then sequenced with a 75 bp single-end read on the Illumina Nextseq500 platform.

Construction of single-cell RNAseq: The 10× Genomics© Chromium system was used to capture 5000 single cells. The sequencing library was generated using the 10× Genomics© Single Cell 3′ Solution kit and subjected to Illumina sequencing (NextSeq 500/550 High Output Kit v2.0 75 Cycles).

Computational methods: RNAseq and MARseq samples were analyzed using the UTAP pipeline[91]. Briefly, sequenced reads were trimmed with cutadapt[92] and were mapped to GRCh37/hg38 reference genome with STAR[93] v2.4.2a with the following parameters alignEndsType EndToEnd, outFilterMismatchNoverLmax 0.05, alignSoftClipAtReferenceEnds No. The MARseq gene quantification of the most 3′ 1000 bp of each gene was performed using HTSeq-count[94] in union mode while marking UMI duplicates (in-house script and HTSeq-count). For the RNA-seq samples, we used the gene count of STAR. The pipeline further applied DESeq2[95] with the parameters: betaPrior=True, cooksCutoff=FALSE, and independentFiltering=FALSE for normalization and testing for differential expression. Raw $p$ values were adjusted for multiple testing using Benjamini and Hochberg procedure[96]. Genes with base mean > 10, log2FC > 1 and padj <0.05 were considered as differentially expressed.

The differential splicing variations were analyzed using the default parameters with MAJIQ[43,97] using the default parameters. Events were considered significantly different using deltaPSI > =0.20, as recommended by the authors.

Functional enrichment was performed with QIAGEN Ingenuity Pathway Analysis IPA (QIAGEN Inc., https://digitalinsights.qiagen.com/IPA)[98], WebGestalt[44], and GeneAnalytics[99].

For the single-cell RNAseq data alignment, quantitation and aggregation of sample count matrices were performed using the 10× Genomics Cell Ranger 2.0.0. 10× Genomics Loupe software was used for data visualization. Libraries prepared from both sexes were included as our data indicated that sex-bias in gene expression was neglectable compared to the genotype effect.

Organotypic slice cultures: *Hnrnpu*$^{fl/fl}$ or ICR mice were mated with identical genotypes and electroporated in utero at E13. The DNA mix injected into the ventricles contained 1.5ug of CAG::Cre-ERT2 and 1 µg/ul of the dual Cre reporter (CAG Spotlight reporter, CAG::SL, carrying a constitutively expressed floxed ZsGreen whose removal allows the expression of a mCherry). One day later (E14), the brains were removed from the embryos and kept in oxygenated cold L15 supplemented with Glucose (0.6%), embedded in 3.5% Agarose, and sliced on a Leica Vibrotome into 300uM thick slices. The slices were placed on membrane inserts (millicell, 0.4 µm, Millipore) and placed on MEM:HBSS (1:1) supplemented with 25% Hours Serum, 4.5 mg/ml Glucose, and Penicillin-Streptomycin. Tamoxifen (1 mM) or DMSO were added to the media. Slices were embedded on a glass-bottom plate in 50% Matrigel: Media for imaging. Imaging was carried out in a spinning disk confocal microscope based on an OLYMPUS IX83 inverted microscope, VisiScope CSU-W1-T1 confocal system (Visitron Systems, Germany), and an sCMOS 4.2 MPixel camera.

In utero electroporation (IUE) and primary cultures: ICR mice at 14 d after gestation were anesthetized with 10% ketamine/20 mg/ml xylazine (1/10 mixture, 0.01 µl/g of body weight, i.p.). The uterine horns were exposed, and plasmids mixed with Fast Green (2 µg/µl; Sigma) were microinjected by mouth pipette through the uterus into the lateral ventricles of embryos by pulled glass capillaries (Sutter Instrument). Electroporation was accomplished by discharging five 35 mV, 50-ms-long pulses with 950 ms intervals generated by a NepaGene electroporator. The pulses were delivered using 10-mm-diameter platinum-plated tweezer electrodes (Protech International) situated at either side of the head of each embryo through the uterus. Animals were killed 4 days after electroporation. Embryos were perfused (4% paraformaldehyde), and their brains were removed. For migration assays, E14 embryos were intraventricularly injected with 6 µg px330 CRISPR/CAS9 or px330 CRISPR/CAS9—sgRNA's (two sequences targeting each gene of

choice and 1 μg of CAG::GFP. For lineage tracing, electroporation was done at E14 with either or control px330 plasmid or CRISPR/CAS9 *Hnrnpu* sgRNAs together with an episomal CAG::GFP and an artificial transposon (pB CAG::DsRed)-transposase (pBASE). For radial migration analysis, four days post IUE, embryos were perfused intracardiacally with cold 4% PFA-PBS; brains were then removed and postfixed ON. Processing of P21 brains was done following transcardial perfusing. After washing, fixed brains were embedded in 3.5% low melting agarose and sectioned to 60 μM thick slices using Leica VT1000 S vibrating blade microtome. The sections were mounted to a microscope slide, stained with DAPI, and mounted in ProLong Gold Antifade Mountant (Invitrogen). To assess the changes in protein expression, brains were removed from the embryos, and GFP expressing cortical tissue was dissected, enzymatically dissociated in a Neural Tissue Dissociation kit (P) using gentleMACS™ Dissociator, Milteny. Dissociated cultures were plated on PLL/laminin-coated glass and kept in culture in MEM, 0.6% Glucose, GlutaMAX (×1), 5% Horse Serum (HS), 5% Fetal calf serum (FCS), and Gentamycin. After 2 days, the primary cultures were fixed, stained with antibodies of choice, and imaged on Leica DMi8 with an Andor Dragonfly 202, spinning disk confocal.

Click chemistry: EdU solution (PBS) was delivered by Intraperitoneal injection to pregnant dames (0.05 mg EdU per g body weight). Half an hour to an hour post-injection (as indicated), embryos were harvested fixed in 4% PFA overnight for further processing. Alternatively, the final concentration of 1 mM of EdU dissolved in DMSO was added to the media in which Neurospheres (NS) were cultured. Cryo-sections or glass adherent NS, were treated with DNase (0.03 U/μl) for 15 min, and blocked for 30 min with blocking solution (PBS 0.1% Triton x-100, 10% Horse Serum (HS), 10% FCS). Sections were later immersed for 30 min in Click reaction mix (PBS containing 100 mM Tris-HCl, pH 8.5, 1 mM CuSo4 2.5 μM Cy3-Azide or Cy5-Azide, and 100 mM Ascorbic acid). Samples were washed before further immunohistochemical analysis.

Additional information can be found in the Supplementary information tables. Supplementary information table 1: primary antibodies list. Supplementary information table 2: secondary antibodies list. Supplementary information table 3: chemical list. Supplementary information table 4: media. Supplementary information table 5: kits. Supplementary information table 6-: oligonucleotides. Supplementary information table 7: recombinant DNA. Supplementary information table 8: softwares.

**Reporting summary**. Further information on research design is available in the Nature Research Reporting Summary linked to this article.

## Data availability
Source data are provided in this paper. The RNA-seq data generated in this study have been deposited in the NCBI GEO database under accession code GSE181527. The additional data generated in this study are provided in the Supplementary Information/ Source Data file. Further information and requests for resources and reagents should be directed to and will be fulfilled by the lead contact, Orly Reiner (orly.reiner@weizmann.ac.il). Source data are provided with this paper.

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

## Acknowledgements

We thank Profs. Moshe Oren and Eli Arama, and Dr. Yael Aylon for helpful discussions, and Dr. Kitty Reemst for her contribution to the project. O.R. is an incumbent of the Bernstein-Mason Chair of Neurochemistry, Head of the M. Judith Ruth Institute for Preclinical Brain Research, and T.S. is incumbent of the Leir Research Fellow Chair in Autism Spectrum Disorder Research. Our research is supported by a research grant from the William and Joan Brodsky Foundation and the Edward F. Anixter Family Foundation, the Helen and Martin Kimmel Institute for Stem Cell Research, the Nella and Leon Benoziyo Center for Neurological Diseases, the David and Fela Shapell Family Center for Genetic Disorders Research, the Brenden-Mann Women's Innovation Impact Fund, the Richard F. Goodman Yale/Weizmann Exchange Program, The Irving B. Harris Fund for New Directions in Brain Research, the Irving Bieber, M.D. and Toby Bieber, M.D. Memorial Research Fund, The Leff Family, Barbara & Roberto Kaminitz, Sergio & Sônia Lozinsky, Debbie Koren, Jack and Lenore Lowenthal, and the Dears Foundation. The research has been supported by grant No. 2397/18 from the Canadian Institutes of Health Research (CIHR), the International Development Research Centre (IDRC), the Israel Science Foundation (ISF), and the Azrieli Foundation, and grant No. 545/21 from the ISF, German-Israeli Foundation (GIF; Grant no. I-1476-203.13/2018), and United States-Israel Binational Science Foundation (BSF; Grant No. 2017006, joint grant to T.S., O.D., D.B.G, and O.R.).

## Author contributions

T.S. led the research, conducted all animal-related experiments described in the manuscript, harvested material for library construction and prepared ×10 Genomics libraries, prepared all the figures, and contributed to the manuscript writing. A.K. constructed MARS-seq libraries and assisted in pipeline analysis of MARS-seq data. A.G. performed all RT experiments designed and performed splice sensitive PCR and western blot analysis. T.O. lead all bioinformatics analyses. Z.P. conducted the ImageStream run and downstream analysis of the FACS data. I.E.S. acquired funds for research, contributed genetic findings regarding HNRNPU mutations, and critically reviewed the manuscript. D.B.G. was involved in funding acquisition and contributed insightful critical remarks, and his lab hosted the bulk-RNA sequencing. O.D. was engaged in funding acquisition, continuously discussed the data, contributed a clinical point of view to the study, and was involved in manuscript writing. O.R. was involved in project conceptualization, funding acquisition, supervision, and manuscript writing.

## Competing interests

D.B.G. is the founder of Praxis Precision Medicines and Action Biosciences. O.D. receives grant support from NINDS, NIMH, MURI, CDC, and NSF. He has equity and/or compensation from the following companies: Tilray, Receptor Life Sciences, Qstate Biosciences, Hitch Biosciences, Tevard Biosciences, Regel Biosciences, Script Biosciences, Actio Biosciences, Empatica, SilverSpike, and California Cannabis Enterprises (CCE). He has received consulting fees from Zogenix, Ultragenyx, BridgeBio, and Marinus. He holds patents for cannabidiol in treating neurological disorders, but these are owned by GW Pharmaceuticals, and he has waived any financial interests. He holds other patents in molecular biology. He is the managing partner of the PhiFund Ventures. Ingrid Scheffer has served on scientific advisory boards for BioMarin, Chiesi, Eisai, Encoded Therapeutics, GlaxoSmithKline, Knopp Biosciences, Nutricia, Rogcon, Takeda Pharmaceuticals, UCB, Xenon Pharmaceuticals; has received speaker honoraria from GlaxoSmithKline, UCB, BioMarin, Biocodex, Chiesi, Liva Nova and Eisai; has received funding for travel from UCB, Biocodex, GlaxoSmithKline, Biomarin and Eisai; has served as an investigator for Anavex Life Sciences, Cerecin Inc, Cereval Therapeutics, Eisai, Encoded Therapeutics, EpiMinder Inc, Epygenyx, ES-Therapeutics, GW Pharma, Marinus, Neurocrine BioSciences, Ovid Therapeutics, Takeda Pharmaceuticals, UCB, Ultragenyx, Xenon Pharmaceutical, Zogenix and Zynerba; and has consulted for Atheneum Partners, Care Beyond Diagnosis, Epilepsy Consortium, Ovid Therapeutics, UCB and Zynerba Pharmaceuticals; and is a Non-Executive Director of Bellberry Ltd and a Director of the Australian Academy of Health and Medical Sciences and the Australian Council of Learned Academies Limited. The other authors declare no competing interests.
