## [Peer Review File · Nature Communications]

Heterogeneous Nuclear Ribonucleoprotein U (HNRNPU)
Safeguards the Developing Mouse CortexREVIEWER COMMENTS

Reviewer #1 (Remarks to the Author):

Sapir et al. reported in this manuscript that the HNRNPU protein is required for embryonic mouse brain neuronal progenitors to survive. The expression and alternative splicing of multiple genes involved in cell survival, cell motility, and synapse formation are affected following Hnrnpu's conditional truncation. They also identified pharmaceutical and genetic agents that can rescue neuronal stem cell death were able to restore neuronal migration. They conclude that their studies point to novel roles of HNRNPU during brain development.

In general, the study was well conducted, and the observations are interesting and potentially important for better understanding the roles of HNRNPU in brain development. However, there is a major concern that needs to be addressed before reaching a clear and solid conclusion.

Major Concern:

1) The authors used Emx1Cre mice to delete/truncate HNRNPU in neuronal stem cells and neurons. However, Emx1 is also expressed in astrocytes and oligodendrocytes, leading to HNRNPU deletion/truncation also in astrocytes and oligodendrocytes. The authors should experimentally test the contributions of potential astrocyte and oligodendrocyte deficits to the observed brain development deficits.

Minor Comments:

1) The authors seemed inter-changeably used neural stem cell versus neuronal stem cell in the manuscript. Since they are different, the authors should be consistent and define clearly what they exactly mean.

2) All n numbers for each experiment should be clearly provided in each figure legend.

Reviewer #2 (Remarks to the Author):

This study by Sapir, Reiner, and colleagues examined the function of the Hnrnpu gene, whose dysregulation is associated with various human disorders. The authors focused on the developmental role of the gene in the cortex and used genetic, molecular, and imaging approaches to address the question. The authors identified the p53 pathway, amongst others, as key molecules affected by HNRNPU truncation. The authors also achieved a partial rescue of HNRNPU truncation by deleting Tp53 or by using various cell death inhibitors. Although the subject is interesting and significant, the evidence presented here has major weaknesses as outline below, which renders the conclusion unconvincing.

Major issues:

1. The cell death of neural progenitors/stem cells in Hnrnpu fl/fl mutant was not definitely shown. The current data were collected using cultured neurons/slices and using reporter proteins. Since the cortex is still present at E18, although much reduced, the authors should be able to provide data from in vivo studies using apoptosis markers. Alternatively, as the authors were able to perform sgRNA treatment using in utero electroporation (IUE), as shown in Figure 6D, the author should also be able to demonstrate cell death in these embryos. In addition, the number of neural progenitors needs to be compared in vivo at different stages (such as by EdU labeling) to explain the progressive loss of stem cells. Similarly, the rescue of the progenitor loss by deleting Tp53 in the Hnrnpu fl/fl background

needs to be demonstrated using the abovementioned *in vivo* assays. The reviewer understands this is significant amount of work, but feels that it is necessary to support the claim that *Hnrnpu* is required for the survival of neural stem cells, but not for postmitotic neurons.

2. In Figure 5C, activated caspase 3 staining was widespread and was not restricted to the VZ/SVZ, arguing against the notion that *Hnrnpu* truncation primarily affects neural progenitor survival. By contrast, in Figure 6D, *in utero* electroporation of *Hnrnpu* sgRNA did not appear to cause significant thinning of the cerebral wall due to the loss of neural progenitors, although the migration of upper layer neurons was visibly reduced. Therefore, it is not clear from the existing data where cell death occurs.

3. It is not specified if the sgRNA treatment against *Hnrnpu* generates the same truncated HNRNPU or eliminates the protein completely. These two genetic conditions are most likely to affect the gene function differentially, as the truncated protein is expected to only disrupt the RNA-binding ability. Along the same line, the authors suggested that the amount of the truncated protein is reduced in the floxed allele in Figure 5. However, western blotting in supp Figure 1F showed that the levels of truncated and full-length proteins were comparable. Therefore, the genetic conditions resulting from Cre-mediated and sgRNA-mediated deletions are not necessarily the same, or even similar.

4. The effects of *Hnrnpu* truncation/deletion on *Mdm2* are not consistent among the figures. For example, Figure 4C showed a reduction in the overall level of *Mdm2*. This was not the case in Figure 3E or Figure 6C. In addition, the alternative splicing patterns for *Mdm2* were different between Figure 3E/Figure 6C and Supplementary Figure 5A.

5. In Figure 6C & D, *Srsf3* and *Hnrnpu* sgRNA treatment appeared to cause similar effects in *Mdm2* alternative splicing (i.e., exon 3 was more skipped; *Srsf3* had a milder effect) and in reducing neuronal migration (*Srsf3* had a stronger effect here). This contradicts the notion that the two splicing factors have opposing effects.

6. In all figures demonstrating the cortex (other than Figure 6D), there was no labeling of the orientation or of the layers, making it very difficult to assess the phenotypes.

7. In Figure 1A, why was EdU seen in superficial layers after 1 hr labeling?

Taken together, *in vivo* studies using the floxed mutant and/or IUE, and detailed temporal and spatial analyses of apoptosis are much needed. They will help demonstrate where cell death occurs *in vivo* as a result of *Hnrnpu* deletion or truncation.

Minor issues:

1. WT1 and WT2 in Figure 5 and Supplementary Figure 5 should be labeled as Control1/Ctrl1 and Control2/Ctrl2, as the animals contained transgenic alleles.

2. The resolution of Supplementary Figure 4 is too low to view.

3. fl/fl was used to indicate the deletion of *Hnrnpu*, but loxP/loxP was used for *Tp53*. It'd be better to be consistent.

4. Supplementary Figure 5 was titled "Rescue of alternative splicing of MDM2 and VZ organization in *Hnrnpu* mutant cortices following *Tp53* KO". However, *Tp53* KO does not alter the AS by itself and also does not restore the AS in *Hnrnpu* fl/fl background. The title is therefore misleading. "E24 WT cortices" in the same legend should be E14.

5. Figure 1F-I, sections were labeled as 5 μ M. Should be 50 μ m?

REVIEWER COMMENTS

Reviewer #1 (Remarks to the Author):

Sapir et al. reported in this manuscript that the HNRNPU protein is required for embryonic mouse brain neuronal progenitors to survive. The expression and alternative splicing of multiple genes involved in cell survival, cell motility, and synapse formation are affected following *Hnrnpu*'s conditional truncation. They also identified pharmaceutical and genetic agents that can rescue neuronal stem cell death were able to restore neuronal migration. They conclude that their studies point to novel roles of HNRNPU during brain development.

In general, the study was well conducted, and the observations are interesting and potentially important for better understanding the roles of HNRNPU in brain development. However, there is a major concern that needs to be addressed before reaching a clear and solid conclusion.

Major Concern:

1) The authors used *Emx1*Cre mice to delete/truncate HNRNPU in neuronal stem cells and neurons. However, *Emx1* is also expressed in astrocytes and oligodendrocytes, leading to HNRNPU deletion/truncation also in astrocytes and oligodendrocytes. The authors should experimentally test the contributions of potential astrocyte and oligodendrocyte deficits to the observed brain development deficits.

We thank the reviewer for raising the issue of non-neuronal cells. To address this issue, we performed several experiments, which we present hereafter. Our observations suggest that the death of *Emx1::Cre* expressing cells that carry two copies of *Hnrnpu* floxed alleles and their progeny is progressive and extensive. We observed that the cell death starts medially and progresses laterally, in a gradient that mirrors the expression pattern of *Emx1*. We first looked at E18 brains from controls, *Hnrnpu*^{+/+} *Emx1*^{Cre/+}, and *Hnrnpu*^{fl/fl} *Emx1*^{Cre/+} littermates. We stained these brains with CM5 (anti-Tp53, Leica) to map the region where p53 is stabilized (Fig. 4 C-F). We used glial fibrillary acidic protein (GFAP) to stain astrocytes that occupy the cortical plate (Fig. 1 J-P). We additionally immunostained the sections using anti-TBR1 antibodies (Fig. 1 K, N). TBR1 is expressed soon after cortical progenitors differentiate and is highly expressed in early-born neurons that occupy the deep cortical layer 6. We noticed that at E18, the medial parts of the cortex of *Hnrnpu*^{fl/fl} *Emx1*^{Cre/+} embryos were lost, and the lateral ventricle seemed to be exposed (Fig. 1J, Fig. 4E). This region is rich in cells with high levels of nuclear TP53. Since more laterally, a structure resembling the cortex is still visible, we determined that the superficial regions of the remaining cortical structure are GFAP positive, yet markedly reduced levels of GFAP staining are detected. The deep layer 6, TBR1 positive cells no longer resided at a deep position but are spread throughout the width of the cortex (Fig. 1 K vs. N). To estimate the reduction in GFAP signal and distribution, we produced intensity histograms from the upper part of the cortex in sections of n=3 brains for each genotype (Fig. 1 P). Both genotypes showed a signal peaking in the basal edge of the cortical plate, yet they were remarkably different. We found that the integration of the signal over 300 microns (area under the curve) was significantly reduced in the mutant compared to control brains (13825±209.1 units in the control sections vs. 8461± 168.4 in the mutant brains, n=3, unpaired t-test, P<0.0001).

In addition, we looked at P21 brains. This time point was the most advanced age in which mutants could be collected as they could not survive after weaning. The entire cortical region is missing at this age, as seen as early as P8 (Fig. 1I, schematically shown next to panel T). P21 brain sections of mutant and control (*Hnrnpu*^{+/+} *Emx1*^{Cre/+}) littermates were stained with deep and superficial cortical plate layers markers (Tbr1, Cux1 Fig 1Q, T) 2',3'-cyclic nucleotide-3'-phosphodiesterase (CNPase) that stains myelinating glia and myelin (Fig. 1R, U) and GFAP (Fig. 1 S, V). Although we could detect at the appendix that persists at the most ventrolateral region of the cortex both GFAP and CNPase and some positive neuronal markers, the area lacks any organization. It could not be assigned an anatomical annotation.

Due to this dramatic phenotype, we chose a subtler approach (Supp. Fig 1). We lineage traced (E14 to P21) cells treated with either *Hnrnpu* sgRNA or control px330 plasmid. To this end, we electroporated E14 embryos with a combination of CAG::GFP and transposase-transposon system containing pCAG::PBASE and a dsRed donor plasmid, pB CAG::DsRed. This experiment allowed us to distinguish lineage traced astrocytes and earlier born neurons (Supp. Fig. 1A-G). The first-born neurons expressed GFP from the episomal plasmid CAG::GFP and dsRed from the piggybac one and resided in layers II-IV. The later-born progeny appeared red as the cells lost the episomal GFP but expressed dsRed from the integrated transposon. These cells exhibited an astrocytic appearance and expressed GFP under a GFAP promoter (data not shown). We used CRISPR/CAS9 constructs targeting the *Hnrnpu* locus as an effective way to reduce HNRNPU levels (see Supp. Fig. 8). We sacrificed the mice (n=3 of each treatment at P21) and calculated the proportion of the dsRed+GFP- out of all labeled cells (single or double-labeled, Supp. Fig 1 H-J). We noticed that the percentage of late-born astrocytes at P21 was similar, regardless of HNRNPU levels. Some ectopic neurons could be detected in the *Hnrnpu* sgRNA's treated brains consistent with migration deficits that we reported in the main text. Taken together, we concluded that the progenitor's death affected all their

progeny, neurons and astrocytes alike, but could not evaluate the relative vulnerability of astrocytes relative to neurons as a result of lower dosage of HNRNPU. The new results are incorporated to Fig 1, Fig 4, and Supp. Fig 1.

We addressed this point in the modified discussion.

Neural progenitors were highly vulnerable to HNRNPU protein depletion, whereas postmitotic neurons *in vitro* and neurons and astrocytes *in vivo* did not display similar sensitivity to HNRNPU loss. This difference was not correlated with the expression pattern since HNRNPU was highly expressed in radial glia as well as in postmitotic neurons and astrocytes.

Minor Comments:

1) The authors seemed interchangeably used neural stem cell versus neuronal stem cell in the manuscript. Since they are different, the authors should be consistent and define clearly what they exactly mean. Following this remark, the term "Neuronal" was changed to "neural," and the term "Stem cells" was changed to "progenitors".

This study was done in the earliest time point of E13- E14 mouse brains, populated by neural progenitor cells (NPCs), the progeny of neuroepithelium division. NPCs either self-proliferate or give rise to intermediate progenitors and neurons and later glial lineages. The text was changed accordingly.

2) All n numbers for each experiment should be clearly provided in each figure legend.

Numbers were added wherever they were missing.

Reviewer #2 (Remarks to the Author):

This study by Sapir, Reiner, and colleagues examined the function of the *Hnrnpu* gene, whose dysregulation is associated with various human disorders. The authors focused on the developmental role of the gene in the cortex and used genetic, molecular, and imaging approaches to address the question. The authors identified the p53 pathway, amongst others, as key molecules affected by HNRNPU truncation. The authors also achieved a partial rescue of HNRNPU truncation by deleting Tp53 or by using various cell death inhibitors. Although the subject is interesting and significant, the evidence presented here has major weaknesses as outline below, which renders the conclusion unconvincing.

Major issues:

1. The cell death of neural progenitors/stem cells in the *Hnrnpu* fl/fl mutant was not definitely shown. The current data were collected using cultured neurons/slices and using reporter proteins. Since the cortex is still present at E18, although much reduced, the authors should be able to provide data from *in vivo* studies using apoptosis markers.

Alternatively, as the authors were able to perform sgRNA treatment using *in utero* electroporation (IUE), as shown in Figure 6D, the author should also be able to demonstrate cell death in these embryos. In addition, the number of neural progenitors needs to be compared *in vivo* at different stages (such as by EdU labeling) to explain the progressive loss of stem cells. Similarly, the rescue of the progenitor loss by deleting Tp53 in the *Hnrnpu* fl/fl background needs to be demonstrated using the abovementioned *in vivo* assays. The reviewer understands this is significant amount of work, but feels that it is necessary to support the claim that *Hnrnpu* is required for the survival of neural stem cells, but not for postmitotic neurons.

We thank the reviewer for these insightful comments. We indeed followed the suggestion to look at the persistent cortex at E18. We compared E18 brains from controls, *Hnrnpu*^{+/+} *Emx1*^{Cre/+}, and *Hnrnpu*^{fl/fl} *Emx1*^{Cre/+} littermates immunostained these brains with CM5 (anti-Tp53, Leica) to map the region where p53 is stabilized (Fig 4 C-F). We could detect Tp53 accumulation in cells throughout the entire width of the remaining cortex. We could not use this data to support or dispute the notion that death occurs in a subset of cells.

Alternatively, as the authors were able to perform sgRNA treatment using *in utero* electroporation (IUE), as shown in Figure 6D, the author should also be able to demonstrate cell death in these embryos.

Based on our *in vitro* data (Fig. 2), we concluded that CAG-driven Cre deletion of *Hnrnpu* floxed alleles causes progenitors' death. At the same time, late activation of Cre (under $T\alpha 1$ promoter) allows some neurons to survive. Similarly, we found that using *in utero* electroporation, the introduction of *Hnrnpu* sgRNA at E14 enables neurons to persist with reduced levels of HNRNPU (Supp. Fig 8). To test the extent of cell death in brains treated with *Hnrnpu* sgRNA, we followed the reviewer's suggestion and electroporated brains with either *Hnrnpu* sgRNAs, *Srsf3* sgRNAs, a combination of both, or control px330. The brains were collected four days post EP, and the sections were immunostained with Cleaved Caspase 3 (CC3). The findings are presented in Supp. Fig. 9, showing sporadic cells that appear to be CC3 positive in both the *Hnrnpu* sgRNAs and *Srsf3* sgRNAs and none in which both *Hnrnpu* and *Srsf3* were knocked out. Yet, due to the small number of cells, we may conclude that *in utero*, cell death that is seen four days after *in utero* introduction of CRISPR/CAS9 *Hnrnpu* sgRNA is minimal, and no strong statements can be made regarding the effect of the different sgRNA treatments.

These new results are shown in Supp. Fig. 9.

In addition, the number of neural progenitors needs to be compared *in vivo* at different stages (such as by EdU labeling) to explain the progressive loss of stem cells.

Based on our data and the E14-18 experiment presented earlier, we claimed that progenitors present higher sensitivity to *Hnrnpu* loss of function. We injected EdU to E14 pregnant mice 30 minutes before sacrificing the female to look at their progressive loss. We immunostained the brain sections with anti-Sox2 antibodies that mark self-renewing neural stem cells and anti-Tbr2 antibodies that mark intermediate progenitors. We focused on the regions immediately adjacent to the extensive cell death at the medial part of the *Emx1* expressing domains. We found that mutant brains' ventricular/subventricular zones (VZ/sVZ) express Sox2 and Tbr2. These areas show a dramatic size reduction and display a marked decrease in EdU incorporation (Supp. Fig. 3). EdU incorporation reduction was measured and presented later in the manuscript (Fig. 5AE); see next paragraph.

These results are now presented in Supp. Fig 3, quantification was included in Fig. 5 AE.

Similarly, the rescue of the progenitor loss by deleting *Tp53* in the *Hnrnpu* fl/fl background needs to be demonstrated using the abovementioned *in vivo* assays. The reviewer understands this is significant amount of work, but feels that it is necessary to support the claim that *Hnrnpu* is required for the survival of neural stem cells, but not for postmitotic neurons.

We followed the reviewer's advice and have used EdU to label S phase cells in the cortical plate of E14 littermates. We obtained three rescued brains (*Hnrnpu*^{fl/fl} *Emx1*^{Cre/+} *Tp53*^{loxP/loxP}). In those mice, we observed rescue of the cortical structure and could count the number of cells in identical areas (100x100 μM^2) the sVZ of coronal sections (14 μM thick) obtained from these brains and three control brains. Control brains were analyzed for the same litters as the rescued brains. We found that the number of EdU cells in the rescued brains was reduced to 31.7% \pm 3.07 of control values (Average \pm SEM, of n=3 brain, 23 data points). These were partially restored to 56.97% \pm 2.65 compared to the control in the rescued brains (Fig. 5AE). We find this number consistent with our estimation of a partial rescue of the cortices by avoiding p35- dependent cell death.

EdU quantification was included in Fig. 5 AE.

2. In Figure 5C, activated caspase 3 staining was widespread and was not restricted to the VZ/SVZ, arguing against the notion that *Hnrnpu* truncation primarily affects neural progenitor survival. By contrast, in Figure 6D, *in utero* electroporation of *Hnrnpu* sgRNA did not appear to cause significant thinning of the cerebral wall due to the loss of neural progenitors, although the migration of upper layer neurons was visibly reduced. Therefore, it is not clear from the existing data where cell death occurs.

We thank the reviewer for this observation. We indeed saw widespread CC3 staining in E14 *Hnrnpu*^{fl/fl} *Emx1*^{Cre/+} brain sections (Fig. 5 C). We continued and confirmed that p53 accumulation appears in cells in the entire width of the dying cortex (Fig. 4 C-F). Additionally, we stained electroporated brains with CC3 and observed merely sporadic cell death four days post IUE (Supp. Fig. 9). The electroporated brains are, in fact, a mosaic of electroporated cells in a wild-type environment. The reduction in HNRNPU levels in the cells is impressive, but there is variability in the electroporated cells (See Supp. Fig. 8). Additionally, the introduction of the CRISPR/CAS9 *Hnrnpu* sgRNA *in utero* is done at E14, and there is some delay until the construct is expressed (we estimate at least 12 h, based on our *in vitro* studies (Fig. 4B), the gene is deleted, and the reduction in HNRNPU levels occurs. These differences between the

two experimental systems make it hard to conclude whether the loss of HNRNPU results in a specific cell-type sensitivity.

We used E14 slices from EdU treated control and mutant embryonic brains to map the progression of cortical tissue loss (Supp. Fig 3). We indeed saw a massive reduction in the VZ/sVZ (Sox2, Tbr2 positive cells). The cortical plate regions are missing in medial areas and sparse more ventrally. Since the CP is the progeny of the VZ/sVZ, this can be a secondary effect. We did observe some early-born layer six neurons in the cortical structures that still exist at E18 but are almost entirely gone at P8 and P21 (Fig 1 H-I and Q-U). We, therefore, believe that the question of which cells are more sensitive to *Hnrnpu* loss-of-function is not conclusive based on the analysis of *Hnrnpu* deleted brains. We suggest that the experiments presented in Figure 2 and Supplementary Fig. 2 are more informative and support the notion of progenitors' death and more considerable sensitivity to HNRNPU loss of function.

3. It is not specified if the sgRNA treatment against *Hnrnpu* generates the same truncated HNRNPU or eliminates the protein completely. These two genetic conditions are most likely to affect the gene function differentially, as the truncated protein is expected to only disrupt the RNA-binding ability. Along the same line, the authors suggested that the amount of the truncated protein is reduced in the floxed allele in Figure 5. However, western blotting in supp Figure 1F showed that the levels of truncated and full-length proteins were comparable. Therefore, the genetic conditions resulting from Cre-mediated and sgRNA-mediated deletions are not necessarily the same, or even similar.

We agree that both treatments are not identical. Based on others that used the same truncated allele (Ye et al., 2015) (Bagchi et al., 2020) we would like to suggest that the deletion of *Hnrnpu* causes loss of function. Moreover, we observed a reduction in anti-HNRNPU reactivity by immunohistochemistry in brains in which the HNRNPU is truncated when we used an antibody that was raised against the N terminus of the protein, suggesting that the truncated product is either present in reduced levels or is unstable (See Figure 5 S vs. U)

We would like to present these images to the reviewer showing slices from E14 control (A) mutant (B) and rescued (C) brains, stained with anti-N-ter HNRNPU, that reacts with the full length and the truncated HNRNPU protein (yellow). Most cells in the mutant (B) and rescue (C) cortex do not express HNRNPU, suggesting that the truncated protein is not stable. The cells that are HNRNPU+ scattered in the rescue and mutant cortices are likely interneurons that are invading the cortex from the HNRNPU+ ganglionic eminence that do not express *Emx1*. Bones and meninges are still visible (and contain HNRNPU+ nuclei) as sections were done on the embryo head without prior removal of the skull.

4. The effects of *Hnrnpu* truncation/deletion on *Mdm2* are not consistent among the figures. For example, Figure 4C showed a reduction in the overall level of *Mdm2*. This was not the case in Figure 3E or Figure 6C. In addition, the alternative splicing patterns for *Mdm2* were different between Figure 3E/Figure 6C and Supplementary Figure 5A.

Total RNA seq (done on mice cortices) provided quantitative data on both expression levels of all splice variants and highlighted dysregulation of splicing events in intronic levels and gene levels. To verify the under/over-representation of specific splice variants, we performed cDNA using splice variant-specific primers. Both experiments show the difference in the representation of the targeted splice variance, yet the PCR shown in Fig. 3E, Fig 6C, and Supp. Fig. 7A is not suitable to measure expression levels. To obtain quantitative data, we co-electroporated cultured progenitors (Neurospheres, NS) with pCAG::GFP and CRISPR/CAS9 targeting *Hnrnpu*, *Srsf3* sgRNA, or both, and compared the data to control NS electroporated with pX330. We performed qPCR on cDNA prepared from each

treatment and measured the relative levels of *Mdm2* splice variants (MDM2 del E3/MDM2). The results are included in Fig. 6D.

5. In Figure 6C & D, *Srsf3* and *Hnrnpu* sgRNA treatment appeared to cause similar effects in *Mdm2* alternative splicing (i.e., exon 3 was more skipped; *Srsf3* had a milder effect) and in reducing neuronal migration (*Srsf3* had a stronger effect here). This contradicts the notion that the two splicing factors have opposing effects.

We cited that the general accepted notion is that "SR (serine/arginine-rich) splicing factors (SRSF) generally oppose HNRPs with exon skipping and exon retention (Busch and Hertel, 2012; Van Nostrand et al., 2020)." However, we agree with the reviewer that in each individual splicing event, the effects will differ. It should be noted that both HNRNPU and SRSF3 are members of large protein families that may exhibit, in some cases, redundant and overlapping activities. More specifically, in the experiments we show, the simultaneous manipulation of both genes results in a functional improvement. We modified the text accordingly.

6. In all figures demonstrating the cortex (other than Figure 6D), there was no labeling of the orientation or of the layers, making it very difficult to assess the phenotypes.

We appreciate this remark as the mutant brains are so disturbed their phenotype may be hard to assess. We made an effort to orient the readers better by adding low magnification images of a vast area of the brain and annotating major structures (Fig. 1B, C, J, Fig. 4 C, E), Schematic representation of the locations of the images (Fig. 1 next to panels Q, T, Fig. 5 B,-D, J-L, S-U) or white lines at the edge of the cortex (Fig 5. C-F). We also reoriented the mutant brains in a dorsal up ventral down orientation and indicated the imaged regions, hoping that this will clarify the data.

7. In Figure 1A, why was EdU seen in superficial layers after 1 hr labeling?

The following publications presented pulse labeling with Thymidine Analogues such as BrdU of E14.5 mouse embryos. Both shorter pulses (30 min) or similar pulse to the one the reviewer raised concern, (1h) show superficial cells that are BrdU positive Panels C, E (Pucilowska et al., 2012), and Panel D (Hou et al., 2012). These cycling cells may belong to the meninges, and they are not included in our analysis (we only refer to cells in the VZ and sVZ of the labeled brains)

<https://www.jneurosci.org/content/32/25/8663.long>

(Pucilowska et al., 2012)

Figure 5.

Loss of ERK2 disrupts basal progenitor frequency and generation, resulting in premature progenitor pool depletion. **b**, the number of intermediate progenitors was analyzed by immunohistochemical analysis with Tbr2 (red) at E14.5 ($p = 0.0080$). **c**, A short BrdU (30 min) pulse was intraperitoneally injected, 30 min before sacrificing the pregnant dam. **c, d**, The number of BrdU+ cells (green) was counted at E14.5. **e, f**, To analyze the frequency of cycling SVZ progenitors, we used a short-pulse BrdU labeling paradigm (green) to immunolabel Tbr2+ (red) progenitors in the S phase, Tbr2+/BrdU+ ($n = 5$; $p = 0.0022$). **g, h**, Basal progenitor generation from apical progenitors was assayed by co-labeling Tbr2+ cells with BrdU 16 h post-BrdU injection, allowing some apical progenitors to migrate into the SVZ and express Tbr2 (red) ($n = 5$; $p = 0.0001$).

https://www.researchgate.net/figure/SOD2-deficiency-results-in-a-significant-reduction-in-the-size-of-the-proliferative-zone_fig7_230798287
(Hou et al., 2012)

Figure 5: SOD2 deficiency results in a significant reduction in the size of the proliferative zone and neural progenitor cell (NPC) proliferation in the embryonic cerebral cortex. (A): Representative confocal images showing Ki67 (green) and Tuj1 (red) immunoreactive cells in brain sections from E14.5 wild-type and SOD2^{-/-} littermate embryos, taken from matched sections at the same level of the frontal cortex. (B): Representative higher magnification confocal images of a 200-µm slab of the middle telencephalon wall as indicated in (A) are shown with Ki67 (green) staining. (C): The combination of Ki67 (proliferation marker) and Tuj1 (a marker of differentiated neurons) delineates the zones of cell proliferation and differentiation (A). The thicknesses of the proliferative zone (Ki67⁺) and differentiation zone (Tuj1⁺) were quantified, and values represent the mean ± SD of analyses performed on brain sections from three WT and three SOD2^{-/-} mice. *, p < .05; **, p < .001. (D): Representative confocal images of a 200-µm slab of the middle telencephalon wall of WT and SOD2^{-/-} littermate mice at E14.5 immunostained with BrdU (green) and PI (red). Pregnant dams at E14.5 were pulse 6-hour labeled with BrdU, and brain sections were immunostained with BrdU (green) and counterstained with PI (red). (E): Total BrdU⁺ cells within a 200-µm slab of the middle telencephalon wall of WT and SOD2^{-/-} littermate mice were quantified. Values represent the mean ± SD of analyses performed on brain sections from three WT and three SOD2^{-/-} mice. **, p < .01. Abbreviations: BrdU, bromodeoxyuridine; PI, propidium iodide; SOD, superoxide dismutase; WT, wild type.

Taken together, in vivo studies using the floxed mutant and/or IUE, and detailed temporal and spatial analyses of apoptosis are much needed. They will help demonstrate where cell death occurs in vivo as a result of Hnrnpu deletion or truncation.

Minor issues:

1. WT1 and WT2 in Figure 5 and Supplementary Figure 5 should be labeled as Control1/Ctrl1 and Control2/Ctrl2, as the animals contained transgenic alleles.

Fig 5, legends, and text were modified as requested.

2. The resolution of Supplementary Figure 4 is too low to view.

We apologize for the low-quality image; it is now replaced with a new figure (Supp. Fig. 6).

3. fl/fl was used to indicate the deletion of *Hnrnpu*, but loxP/loxP was used for Tp53. It'd be better to be consistent.

The nomenclature we use follows the nomenclature of the original papers that created these mice lines with minor modifications. (Ye et al., 2015) describe the line as: "control (*Hnrnpu*^{fl/fl})". We used the abbreviation fl for floxed. (Marino et al., 2000) refer to the conditional p53 mice as "*p53*^{LoxP/LoxP}". Here, we used the gene name *Tp53* but left the loxP nomenclature as is.

As several labs shared both these lines, we felt that this would be easier for the scientific community to place them in the context of the previously published manuscripts.

4. Supplementary Figure 5 was titled "Rescue of alternative splicing of MDM2 and VZ organization in *Hnrnpu* mutant cortices following Tp53 KO". However, Tp53 KO does not alter the AS by itself and also does not restore the AS in *Hnrnpu* fl/fl background.

We thank the reviewer for noting this. Indeed, Tp53 KO prevents cell death but does not alter the splicing of MDM2. The title of the figure (Now Supp. Fig. 7) was corrected accordingly.

The title is therefore misleading. "E24 WT cortices" in the same legend should be E14.

Corrected.

5. Figure 1F-I, sections were labeled as 5 μM. Should be 50 μm?

The labels in the Figures are correct. The size of the bar in panel A' is approximately the size of one cell nucleus. The size of the bars in G and I are in mm.

References

- Bagchi, D., Mason, B.D., Baldino, K., Li, B., Lee, E.J., Zhang, Y., Chu, L.K., El Raheb, S., Sinha, I., and Nepl, R.L. (2020). Adult-Onset Myopathy with Constitutive Activation of Akt following the Loss of hnRNP-U. *iScience* 23, 101319.
- Busch, A., and Hertel, K.J. (2012). Evolution of SR protein and hnRNP splicing regulatory factors. *Wiley Interdiscip Rev RNA* 3, 1-12.
- Hou, Y.Y., Toh, M.T., and Wang, X. (2012). NBS1 deficiency promotes genome instability by affecting DNA damage signaling pathway and impairing telomere integrity. *Cell Biochem Funct* 30, 233-242.
- Marino, S., Vooijs, M., van Der Gulden, H., Jonkers, J., and Berns, A. (2000). Induction of medulloblastomas in p53-null mutant mice by somatic inactivation of Rb in the external granular layer cells of the cerebellum. *Genes Dev* 14, 994-1004.
- Pucilowska, J., Puzerey, P.A., Karlo, J.C., Galan, R.F., and Landreth, G.E. (2012). Disrupted ERK signaling during cortical development leads to abnormal progenitor proliferation, neuronal and network excitability and behavior, modeling human neuro-cardio-facial-cutaneous and related syndromes. *The Journal of neuroscience : the official journal of the Society for Neuroscience* 32, 8663-8677.

Van Nostrand, E.L., Freese, P., Pratt, G.A., Wang, X., Wei, X., Xiao, R., Blue, S.M., Chen, J.Y., Cody, N.A.L., Dominguez, D., *et al.* (2020). A large-scale binding and functional map of human RNA-binding proteins. *Nature* 583, 711-719.

Ye, J., Beetz, N., O'Keeffe, S., Tapia, J.C., Macpherson, L., Chen, W.V., Bassel-Duby, R., Olson, E.N., and Maniatis, T. (2015). hnRNP U protein is required for normal pre-mRNA splicing and postnatal heart development and function. *Proceedings of the National Academy of Sciences of the United States of America* 112, E3020-3029.

REVIEWER COMMENTS

Reviewer #1 (Remarks to the Author):

The authors have experimentally addressed my major concern and modified the manuscript accordingly. The authors also well addressed my minor comments. This reviewer has no further comment.

Reviewer #2 (Remarks to the Author):

The revised manuscript by Sapir, Reiner, and colleagues addressed some of the major issues. However, the current data still did not provide sufficient support for the key conclusion that Hnrnpu deficiency leads to elevated apoptosis primarily in neural stem cells/progenitors. In fact, the in vivo data showed that apoptosis occurred in progenitors as well as in postmitotic neurons (see below).

Major issues:

1. Apoptosis appeared to occur in neural progenitors as well as in post-mitotic neurons. 1) In Figure 4C-F, Tp53 was stabilized throughout the remaining cortex in the Hnrnpu mutant. 2) Cleaved Caspase 3 (CC3) staining was widespread in Figure 5C. 3) In the rescue experiment shown in Figure 5, cKO of Tp53 using Emx1-Cre not only deleted the gene in neural progenitors, but also in post-mitotic neurons. 4) In supp Fig.9, after Hnrnpu knockdown by sgRNA treatment, sporadic CC3 staining was observed in different regions, not just in the VZ/SVZ. Taken together, none of the in vivo data supports the conclusion that apoptosis occurs predominantly in neural progenitors after loss of Hnrnpu. In fact, the data indicated that cell death occurs in post-mitotic neurons as well. The authors believed that the loss of post-mitotic neurons must be a secondary effect of loss of progenitors. However, this notion is unsubstantiated and is in fact argued against by the in vivo data.

2. Although SRSF and HNRPs generally oppose each other in regulating alternative splicing, the data in this study showed that Srsf3 and Hnrnpu sgRNA treatment caused similar effects in Mdm2 alternative splicing (i.e., exon 3 was more skipped; Srsf3 had a milder effect) and in reducing neuronal migration (Srsf3 had a stronger effect here). Thus, it is inappropriate to conclude that the two splicing factors have opposing effects.

An alternative explanation for the “rescued” effects when combining the two sgRNAs is that each of the two sgRNAs that can enter the electroporated neurons is only half the quantity when only one sgRNA is used. In other words, there is a maximum amount of sgRNAs that neurons can take in, and combining two sgRNAs thus dilutes the effect of each sgRNA.

Minor issues:

The new Supp Fig. 3 is missing the genotypes and the new Supp Fig. 7 is missing the scale bars.

Reviewer #1 (Remarks to the Author):

The authors have experimentally addressed my major concern and modified the manuscript accordingly. The authors also well addressed my minor comments. This reviewer has no further comment.

We thank the reviewer for improving our manuscript.

Reviewer #2 (Remarks to the Author):

The revised manuscript by Sapir, Reiner, and colleagues addressed some of the major issues. However, the current data still did not provide sufficient support for the key conclusion that *Hnrnpu* deficiency leads to elevated apoptosis, primarily in neural stem cells/progenitors. In fact, the in vivo data showed that apoptosis occurred in progenitors as well as in postmitotic neurons (see below).

We thank the reviewer for providing important insights and improving our manuscript.

Major issues:

1. Apoptosis appeared to occur in neural progenitors as well as in postmitotic neurons. 1) In Figure 4C-F, Tp53 was stabilized throughout the remaining cortex in the *Hnrnpu* mutant. 2) Cleaved Caspase 3 (CC3) staining was widespread in Figure 5C. 3) In the rescue experiment shown in Figure 5, cKO of Tp53 using *Emx1-Cre* not only deleted the gene in neural progenitors, but also in postmitotic neurons. 4) In supp Fig.9, after *Hnrnpu* knockdown by sgRNA treatment, sporadic CC3 staining was observed in different regions, not just in the VZ/SVZ. Taken together, none of the in vivo data supports the conclusion that apoptosis occurs predominantly in neural progenitors after loss of *Hnrnpu*. In fact, the data indicated that cell death occurs in postmitotic neurons as well. The authors believed that the loss of postmitotic neurons must be a secondary effect of loss of progenitors. However, this notion is unsubstantiated and is in fact argued against by the in vivo data.

We agree with the reviewer. We indeed detected massive cell death upon *Hnrnpu* loss of function. However, as our data demonstrate, cell death dynamics likely differ in the progenitors and postmitotic neurons. We modified the manuscript's title and the discussion accordingly, following the reviewer's comment.

2. Although SRSF and HNRPs generally oppose each other in regulating alternative splicing, the data in this study showed that *Srsf3* and *Hnrnpu* sgRNA treatment caused similar effects in *Mdm2* alternative splicing (i.e., exon 3 was more skipped; *Srsf3* had a milder effect) and in reducing neuronal migration (*Srsf3* had a stronger effect here). Thus, it is inappropriate to conclude that the two splicing factors have opposing effects.

This is correct. We chose to focus on SRSF3, expecting to find a bias towards MDM2 exon3 retention, as this was our working hypothesis fitting the general notion in the field that SRSF proteins work opposed to HNRNP proteins. " SR (serine/arginine-rich) splicing factors (SRSF) generally oppose HNRPs with exon skipping and exon retention^{54, 55}. " In fact, we observed that *Hnrnpu* and *Srsf3* co-reduction indeed reverses the outcome of *Hnrnpu* loss of function, namely, partially reversed cell death and migration delay. We also noticed that this co-reduction allows to change the ratio between retained and excluded Ex3 variants of MDM2 to resemble the ratios that were observed in the control. However, we concur with the reviewer that we cannot attribute this to the direct splicing activity of *Srsf3* on MDM2. Nevertheless, we could still correlate MDM2 alternative splicing to cell viability and somewhat overcome the mal effects of *Hnrnpu* loss of function.

To avoid confusion, we modified two sentences in the manuscript:

1. Reduced SRSF3 protein level in the absence of HNRNPU restored the ratio of *Mdm2* exon 3 inclusion to wildtype levels and reverted the size of the neurospheres (Fig. 6A-D).

2. The expression and splicing of multiple genes are likely affected following the reduction of HNRNPU and SRSF3. At present, we do not have a comprehensive molecular analysis to explain the observed rescue phenotypes fully.

An alternative explanation for the "rescued" effects when combining the two sgRNAs is that each of the two sgRNAs that can enter the electroporated neurons is only half the quantity when only one sgRNA is used. In other words, there is a maximum amount of sgRNAs that neurons can take in, and combining two sgRNAs thus

dilutes the effect of each sgRNA.

To address this concern, we chose to stain electroporated brain sections with either anti-HNRNPU or SRSF3 antibodies. The sections (70 μ M thick) were taken from two brains of each treatment, and consecutive sections were stained with one or the other antibodies (HNRNPU or SRSF3). We then analyzed the intensity of HNRNPU or SRSF3 staining in the electroporated cells (green cells n=20) and non-transfected cells in their vicinity (n=20). We analyzed n=40 electroporated and n=40 non-transfected cells from brains treated with a combination of sgRNAs. The distribution of the measured intensities is wide, with some cells showing no apparent signal reduction compared to non-transfected cells and others displaying a nearly complete loss of the targeted proteins. We measured the mean intensity in the nucleus (Imaris dots) of the electroporated cells. We calculated it as % of the mean intensity of the non-transfected cells (Signal intensity, % of control). We plotted the data and analyzed it statistically (Supplementary Fig. 10J). We found that either HNRNPU or SRSF3 signal reduction was comparable, regardless of whether it was done separately or in combination with the other sgRNAs. In the new Supplementary Fig 10A-H, we present processed images in which green cells are seen as transparent volumes, and all signal outside of this volume is masked. We highlighted cells with high expression and those that displayed low expression in the same field (yellow and white arrows, respectively)

Supplementary Figure 10: Effective reduction of the expression of target proteins. E14 mouse embryos were electroporated with pCAG::GFP and either CRISPR/CAS9 (px330, A-B) *Srsf3* sgRNA (C-D) *Hnrnpu* sgRNA (E-F) or a combination of the two (G-H). Sections were stained against HNRNPU (A,C,E,G, Red) or SRSF3 (B,D,F,H, Red), and the signal outside the cell volume (Green) was masked. I) Schematic description of the workflow starting with electroporation up to Image analysis. J) Relative intensity of HNRNPU or SRSF3 staining in the electroporated cells (green cells n=20) and non-transfected cells in their vicinity (n=20). In dual sgRNA treatments, n=40 electroporated and n=40 non-transfected cells were measured. Dots represent single measurements. One way ANOVA-ns- non significant, **** p <0.0001. Highlighted are random cells in the field with high expression (yellow) or low expression levels (white). Size bar (A) is indicated (μ M).

Minor issues:

The new Supp Fig. 3 is missing the genotypes, and the new Supp Fig. 7 is missing the scale bars.

Both issues were corrected.

REVIEWERS' COMMENTS

Reviewer #2 (Remarks to the Author):

In the rebuttal letter, the authors agreed that their data did not support the conclusion that cell death occurs predominantly in neural progenitors. On the contrary, their data clearly showed that Hnrnpu is strongly expressed in post-mitotic neurons in the cortical plate (Figure 1) and that Hnrnpu deficiency led to cell death in post-mitotic neurons in vivo (as summarized in the previous review). The recent modifications to the manuscript, in the reviewer's opinion, were minimal to reflect this major change. The reviewer strongly suggests that the authors make consistent changes throughout the manuscript to make their observations and conclusions more coherent.

Minor issues:

There are some spelling and grammatical errors (e.g., HNRNPs were spelled as HNRPs). The main title and some subtitles e.g., "P53 mediates cell death", are overly vague and do not convey clear and specific messages about this study. The manuscript would benefit from professional editing to make it more accurate and polished.

Reviewer #2 (Remarks to the Author):

We thank the reviewer for the time and the efforts to improve our manuscript.

In the rebuttal letter, the authors agreed that their data did not support the conclusion that cell death occurs predominantly in neural progenitors. On the contrary, their data clearly showed that *Hnrnpu* is strongly expressed in postmitotic neurons in the cortical plate (Figure 1) and that *Hnrnpu* deficiency led to cell death in postmitotic neurons in vivo (as summarized in the previous review). The recent modifications to the manuscript, in the reviewer's opinion, were minimal to reflect this major change. The reviewer strongly suggests that the authors make consistent changes throughout the manuscript to make their observations and conclusions more coherent.

To address this issue more accurately, we added more modifications hoping that this issue is better conveyed. Please note that the last revision included a new title to the manuscript that will concur with the reviewer's concern: the original title: "The Heterogeneous Nuclear Ribonucleoprotein U (HNRNPU) is the Safeguard of Neural Stem Cell Viability" was modified to: "Heterogeneous Nuclear Ribonucleoprotein U (HNRNPU) Safeguards the Developing Mouse Cortex". Additionally, we made the following changes:

In the abstract:

Our work revealed that HNRNPU loss of function leads to rapid cell death of both postmitotic neurons and neural progenitors, with an apparent higher sensitivity of the latter.

Finally, we identified pharmaceutical and genetic agents that can partially reverse the loss of cortical structures in *Hnrnpu* mutated embryonic brains, ameliorate radial neuronal migration defects and rescue cultured neural progenitors' cell death.

In the introduction

1. We added a section that describes developmentally programmed cell death in the cortex in the embryonic stage (progenitors) and postnatally (neurons and postmitotic cells).

2. We added a clarification that data regarding sensitivity was gathered in culture:

"Our work revealed that HNRNPU is required for the survival of cultured neural progenitors in vitro and the entire mouse cortex in vivo."

In the results section:

We clarified that postmitotic cells did die following *Hnrnpu* truncation:

"A higher proportion of viable cells expressing excised Cre-reporter was seen when the Ta1 promoter was used (Fig. 2V). **Despite an attenuated death dynamic, postmitotic cells were sensitive to *Hnrnpu* loss of function.**"

The following sentence:

Since *Hnrnpu*'s mutation causes progenitor cell death, we examined whether culturing cortical embryonic neural progenitors as primary neurospheres can recapitulate this phenotype.

Was modified to:

Since *Hnrnpu*'s mutation causes cell death, we examined whether culturing cortical embryonic neural progenitors as primary neurospheres can recapitulate this phenotype.

In the discussion:

1. We again emphasized that postmitotic neurons are sensitive to HNRNPU LOF:

"Neural progenitors and postmitotic neurons were vulnerable to HNRNPU protein. Nevertheless, our data do not rule out the possibility that a broader range of cell types contributes to the dramatic cortical loss following Cre activation in Emx1 expressing cells."

2. We articulated our interpretation of the results in a way that leaves postmitotic cell death as a valid option.

"Based on the different cell death dynamics that we observed *in vitro* using CAG, Tbr2 or Ta promoter-driven HNRNPU truncation, we postulate that cell-specific sensitivity to HNRNPU loss of function may exist."

3. When discussing the sensitivity of progenitors vs. postmitotic neurons, we emphasized that this is based on studies *in vitro*:

"We then focused on the role of P53 in the death of cultured progenitors and on the complete loss of cortical structures *in vivo*."

Minor issues:

There are some spelling and grammatical errors (e.g., HNRNPs were spelled as HNRPs). The main title and some subtitles e.g., "P53 mediates cell death", are overly vague and do not convey clear and specific messages about this study. The manuscript would benefit from professional editing to make it more accurate and polished.

Thank you for these helpful remarks.

The spelling mistake was corrected.

The following titles were modified:

"P53 mediated cell death"

It was changed into:

"HNRNPU loss of function activates P53 mediated cell death"

"Genetic means to rescue Hnrnpu deficiency

Tp53-mediated partial rescue."

It was changed into:

Genetic means to rescue Hnrnpu deficiency *in vitro* and *in vivo*